# Aluminous phyllosilicates promote exceptional nanoscale preservation of biogeochemical heterogeneities in Archaean siliciclastic microbial mats

Keyron Hickman-Lewis [1,2] ✉, Javier Cuadros[3], Keewook Yi [4], Tae Eun Hong[5], Mirang Byeon[5], Jae Hyuck Jang[6], Min-Yeong Choi[6,7], YoonKyung Seo[6,8], Jens Najorka [3], Wren Montgomery [3], Krzysztof Matlak[9], Barbara Wolanin [9], Caroline L. Smith [3,10] & Barbara Cavalazzi[11,12]

Exceptional preservation of biogeochemical complexity in the Precambrian is largely limited to cherts, phosphates and shales; however, ancient fossils, including microbial mats and microbially induced sedimentary structures, also occur, more rarely, in poorly sorted, coarse-grained siliciclastics. The precise micromechanics by which exceptional retention of organic microbial traces occur within such rocks over billion-year geological timescales remain poorly understood. Herein, we explore the micro–nano-scale characteristics of microbial mats preserved in ~2.9 billion-year-old sandstones from the Mosquito Creek Formation (Pilbara, Australia) using a suite of advanced spatially correlated microscopy and geochemistry techniques. We demonstrate that sedimentary horizons rich in K–Al-phyllosilicates exhibit exceptional and unexpected preservation of biogeochemical complexity despite the age and metamorphic grade of the sequence. We propose that authigenic phyllosilicates intercalated with microbial kerogen at the nanoscale promote the preservation of nanoscopic domains of poorly ordered amorphous and turbostratic carbonaceous materials through pressure compensation associated with the kaolinite–illite transition during burial diagenesis and metamorphism, impeding the maturation of organic materials. Elucidating organic preservation in coarse-grained siliciclastics opens new avenues for biosignature searches both in ancient Earth sequences and on Mars, where similar phyllosilicate-bearing sandstones have been collected by the Mars 2020 *Perseverance* rover for near-future sample return.

The Precambrian morphological fossil record, comprising cellular microfossils, microbial mats, microbialites and microbially induced sedimentary structures, provides direct, though limited, evidence for major early steps in biological evolution[1,2]. Its variable preservation and post-depositional alteration has led to numerous ambiguities and controversies in interpretation[2–4]. Most Precambrian organic biosignatures occur in fossil *Konservat-Lagerstätten* within rapidly permineralising micro- and cryptocrystalline cherts and phosphates, or shales that

promote anoxic palaeodepositional microenvironments[3,5,6]. An important but understudied repository of fossilised microbial biomass throughout Earth history has also been identified in coarser-grained siliciclastic sedimentary rocks[7–9]. Bulk geochemical analyses of carbonaceous materials from such horizons suggest that coarse-grained or poorly sorted sedimentary rocks have an unexplored potential to record biogeochemical heterogeneities over billions of years: carbon and nitrogen isotope fractionations in microbial mats from the 3.22 Ga Moodies Group sandstones indicate environmentally distinct signals consistent with autotrophic carbon fixation via the Calvin-Benson and Wood-Ljungdahl pathways, coupled with partial nitrification[10]; carbon and sulphur isotope fractionations of microbially induced sedimentary structures in fine sandstones of the 2.9 Ga Mozaan Group have been interpreted to reflect an early onset of localised oxygenic photosynthesis[11,12]; and the carbon and nitrogen isotope geochemistry of kerogenous laminations in siltstone and sandstones from the 3.0 Ga Lalla Rookh Sandstone suggests methanogenesis harnessing the acetyl CoA pathway and $N_2$ fixation based on FeMo-nitrogenase[13].

The preservation of such biogeochemical heterogeneity in poorly sorted, coarse-grained siliciclastics is unexpected for two reasons: i) their energetic environments of deposition tend to prevent the deposition and retention of low-density organic materials, and ii) their relatively open diagenetic behaviour tends to promote ample diffusion of oxidative fluids and the flushing out of deposited organic materials[14]. Understanding the preservation of microbial biomass in coarse-grained sedimentary sequences therefore requires an understanding of the lithological mechanics promoting organic preservation against these detrimental taphonomic factors. Herein, we elucidate the nanoscale preservational mechanism of primary biogeochemistry within hitherto undescribed microbial mats from sandstones of the Mosquito Creek Formation (MCF), Pilbara, Western Australia.

Samples of MCF sandstones were collected by the Archaean Biosphere Drilling Project (ABDP core #5) through the Mosquito Creek Basin (21°41′53.7″S, 120°37′14.5″E[15]; Fig. S1). A part of the De Grey Supergroup, the age of the MCF horizons cored is constrained between $2971 \pm 15$ Ma and $2905 \pm 9$ Ma[15,16]. The MCF is dominated by poorly to moderately sorted and chemically immature medium- to coarse-grained sandstones (lithic wacke and lithic sandstone) interbedded with shales, wackes, siltstones and conglomerates[15,17,18]. The sequence retains primary sedimentary fabrics and geochemical characteristics; however, it has been influenced by at least five separate deformation events since deposition, during which tight folding and thrust faulting metamorphosed the MCF to sub-greenschist to greenschist grade[15,19,20]. Combining evidence from stratigraphy, sedimentary structures and elemental geochemistry, the palaeodepositional environment has been interpreted as either a continental margin basin connected to the open ocean[15,17,21] or an intracontinental intermontane basin with possible intermittent connection to the open ocean[20].

## Results
### Petrological context of organic materials
Samples were studied using optical microscopy, micro X-ray diffraction (micro-XRD), scanning electron microscopy coupled with energy-dispersive X-ray spectroscopy (SEM-EDS), and Raman and Fourier transform infra-red (FTIR) microspectroscopy to identify and characterise mineralogy and establish the petrological context of relatively organic-rich layers within the samples. MCF sandstones are poorly–moderately sorted and chemically immature arkoses, with grain sizes dominantly between fine–medium silt to coarse sand (<10–500 μm). Matrix phases include major quartz and albite, minor muscovite and dolomite, and rare microcline, andesine, pyrite, anatase, rutile, hornblende, chalcopyrite, chlorite and sodalite (Figs. 1–2, S2–S22). Organic materials occur within discrete laminations and rarely as dispersed fragments (Fig. 1, S2–S6). Coupled micro-XRD, EDS, and Raman and FTIR spectroscopic mapping and point spectral analyses show that relatively organic-rich laminations within the samples occur in particularly close association with aluminous phyllosilicates, which are heavily concentrated in these organic layers but largely absent from the bulk matrix (Figs. 2–3, S7–S22). Aluminous phyllosilicates identified using high-magnification SEM imaging include large, thick plates of muscovite and highly structurally ordered illite with low concentrations of Mg and Fe (platy sub-μm- to μm-scale particles), as well as small amounts of kaolinite, all of which are sub-aligned with the carbon-rich laminations of the microbial mats (Fig. 4D–F). Kaolinite was detected using micro-XRD (Figs. S15–S18) but no pure kaolinite particles were identified with EDS (Figs. S12–S14), suggesting either close contact and intermixing with illite or incomplete transformation of kaolinite to illite. In the bulk matrix, and to a lesser extent within microbial mat layers, K–Al-bearing illite occurs as irregularly oriented domains of 0.1–10 μm platelets, possibly indicating illitisation of kaolinite and feldspars (Fig. 4D–F) rather than allochthonous deposition. Pyrite and rutile occur distributed within the relatively organic-rich layers of the samples and, to a lesser extent, the matrix (Figs. 2A, S12–S20). Very large (50–250 μm) particles of allogenic chlorite and/or muscovite are also present (Figs. S9–S10).

Organic layers are mostly 0.1–0.5 mm in thickness but in exceptional circumstances exceed 1 mm (Figs. 1, S2–S6). They comprise interwoven undulatory kerogenous laminae <30 μm in thickness that are intercalated with layers of sediment particles (e.g. Fig. 1A, B), mantle primary sedimentary textures (e.g. Fig. 1C), and bifurcate/anastomose around sediment grains (e.g. Figure 1D–F). Some organic-bearing layers are very poor in organic materials, comprising thinner, diffuse and discontinuous wisps of kerogen, occurring mostly as an interstitial phase surrounding grains (e.g. Fig. 1F, H). Organic layers also occur at sedimentary interfaces, where they form thick layers with visible internal laminae and entrained sedimentary particles (Fig. 1E). They deform beneath overlying sediment particles and exhibit local rip-ups, tear-ups and crumpling. Sedimentary particles oriented parallel to laminae are common (Fig. 1D–E). Using a focussed ion beam, six ultrathin (<80 nm) sections were extracted from microbial mat horizons for transmission electron microscopy (TEM) and scanning TEM with electron energy loss spectroscopy (STEM-EELS) analyses (Figs. 3, S24–S26). High-angle annular dark field (HAADF) STEM images show that mat laminations comprise discontinuous fragments of anastomosing carbonaceous materials up to 1–2 μm in length and less than 100 nm in thickness, forming a distinct layered fabric intercalated with nanometric–micrometric elongate minerals (Fig. 5). On the basis of the characteristics described above, their sedimentary context, organic composition and morphological identicality to known similar structures from throughout the geological record (e.g refs. 22–24 and references therein), we interpret these laminated structures as fossilised siliciclastic microbial mats.

Potential non-biological alternative explanations for these organic laminations, including layer-parallel secondary hydrocarbon migration along sedimentary layers and non-biological organic concentration after hydrothermal synthesis, are unlikely for three reasons. 1) At the centimetre–nanometre scales of observation reported herein, the organic materials exhibit no characteristics resembling the nanoporous pyrobitumens reported in oil-saturated cherts (e.g refs. 25,26); in particular, there is no petrographic evidence for the emplacement and thermal alteration (coking) of pyrobitumen or its intergrowth with polymetallic sulfides, which has been shown to be diagnostic of such hydrocarbon migration processes in ancient rocks[25]. 2) Throughout the entirety of the cored Mosquito Creek Formation[15,19], no evidence has been found for carbon-rich sediments forming through precipitation from silica-rich, carbon-bearing hydrothermal fluids in vein systems and vent-proximal seafloor sediments that might percolate secondarily through sedimentary laminations (cf[27].). 3) No

geochemical evidence (e.g. strongly positive Eu anomalies or specific trace metal and rare earth element enrichment) exists in these sequences to suggest that hydrothermal processes should have played a dominant–or even significant–role in the generation of the organic carbon studied (cf[15].).

Raman microspectroscopy was used to map the distribution of carbonaceous materials and mineral phases in the aforementioned microbial mat-bearing horizons (Fig. 2A). Consistent with micro-XRD results (Figs. S19–S20), in all samples, the matrix is dominated by quartz and carbonate. Mat laminations are characterised by peaks at ~1340 cm⁻¹ (carbon D peak) and ~1600 cm⁻¹ (carbon G peak). Rutile and phyllosilicates (interpreted from the OH peak at ~3600 cm⁻¹) occur in close association with mat laminations. The thermal maturities of several representative samples were calculated by applying well-established carbonaceous materials-based geothermometers[28,29]. Peak temperatures

were determined for carbonaceous materials within the microbial mat layers and for rare carbonaceous materials within the matrix to evaluate the effects of the presence of phyllosilicates on the peak thermal history of the samples (spectral deconvolution and uncertainties shown in Fig. S23 and Table S1). In general, organic materials in phyllosilicate-rich layers witnessed slightly lower peak temperatures (313–363°C; median = 330°C, mean = 328.5°C) than organic materials within the matrix (314–389°C; median = 345°C, mean = 347.4°C).

## Composition of organic materials

FTIR microspectroscopy was used to investigate the composition and distribution of aromatic and aliphatic moieties within carbonaceous materials (Figs. 2B and 3). The aromatic C=C stretch at 1600 cm⁻¹ (Fig. 3B) and aliphatic CH₂ and CH₃ stretches and C–H bonding at ~2850 cm⁻¹ (weak shoulder on the 2870 cm⁻¹ peak), ~2870 cm⁻¹,

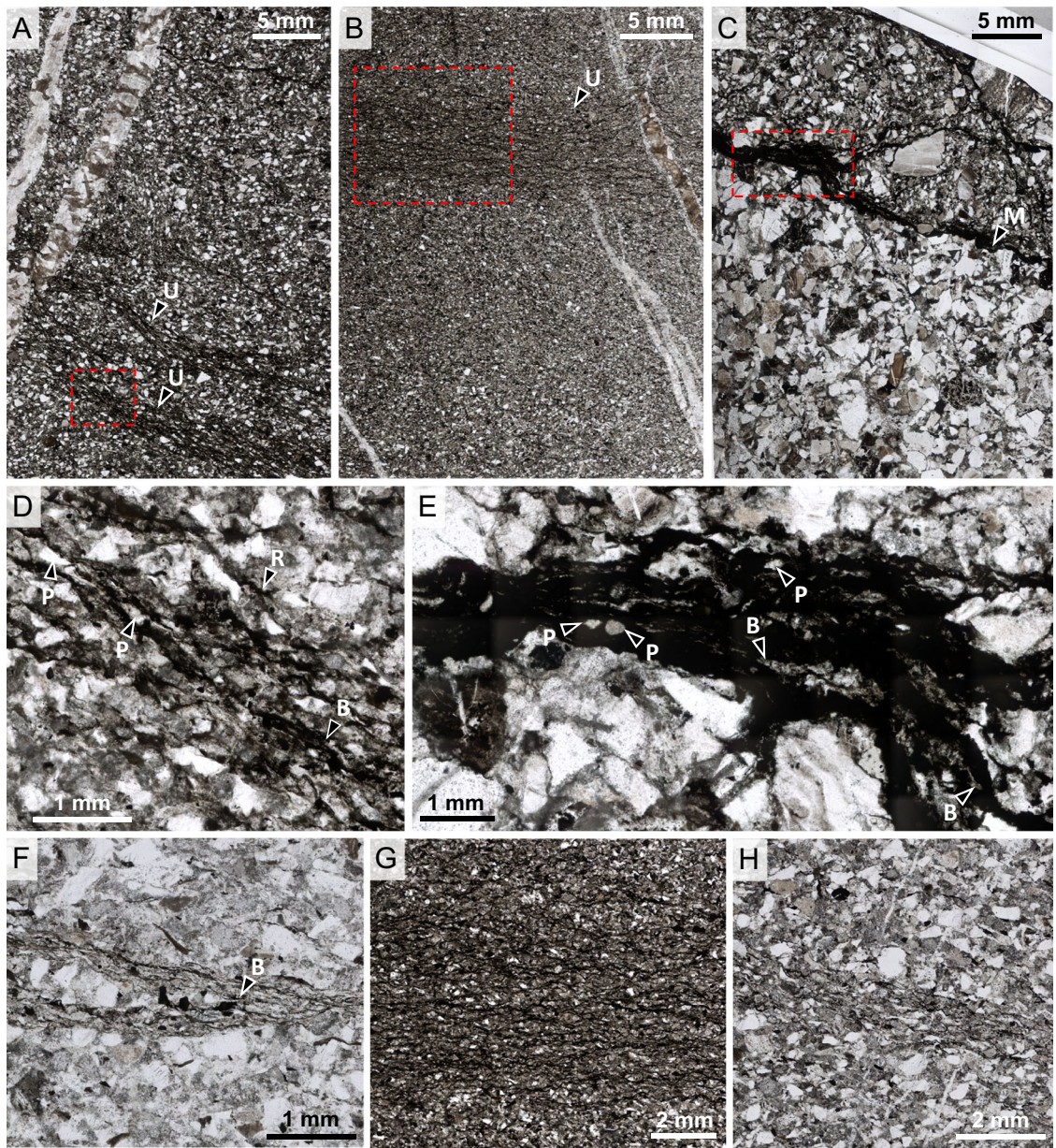

**Fig. 1 | Petrological context and microspectroscopic characterisation of Mosquito Creek Formation microbial mats.** Optical photomicrographs showing organic-rich microbial mat horizons (dark layers) within fine–coarse poorly–moderately sorted arkose. The following microbial mat microstructures are annotated: U = undulatory laminations; M = microbial mat mantling underlying sedimentary layers; P = entrained particles; B = bifurcation and anastomosis of microbial laminations; and R = roll-up. Red box in **A** denotes region detailed in **D**; red box in **B** denotes region detailed in **G**; red box in **C** denotes region detailed in **E**. Samples shown: **A**, **D**) 21b; **B**, **G**) 26 d; **C**, **E**) 28c; **F**) 21a; **H**) 21c.

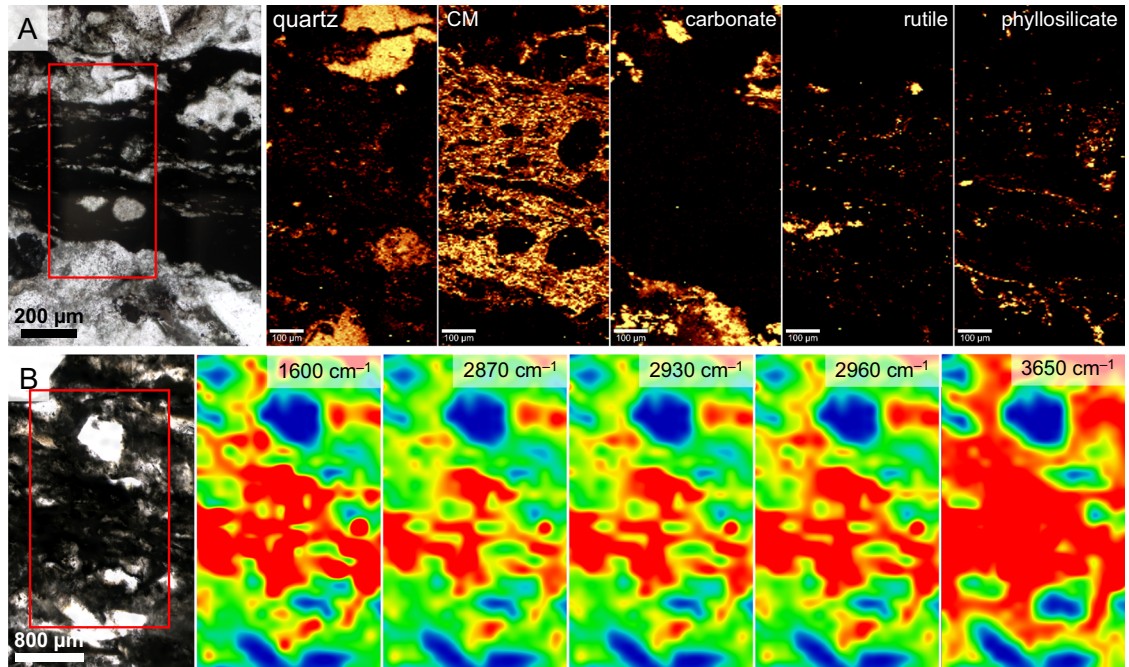

**Fig. 2 | Microspectroscopy characterisation of Mosquito Creek Formation microbial mats. A** 2D Raman phase maps for quartz (465 cm⁻¹), carbonaceous materials (CM; 1600 cm⁻¹), carbonate (1090 cm⁻¹), rutile (610 cm⁻¹) and phyllosilicates (mapped as the O–H band; 3620 cm⁻¹). **B** Fourier transform infrared spectroscopy (FTIR) maps for aromatic carbon (C=C; 1600 cm⁻¹) aliphatic carbon (CH₂ and CH₃; 2870, 2930, 2960 cm⁻¹) and phyllosilicates (mapped as the 3650 cm⁻¹ O–H stretch).

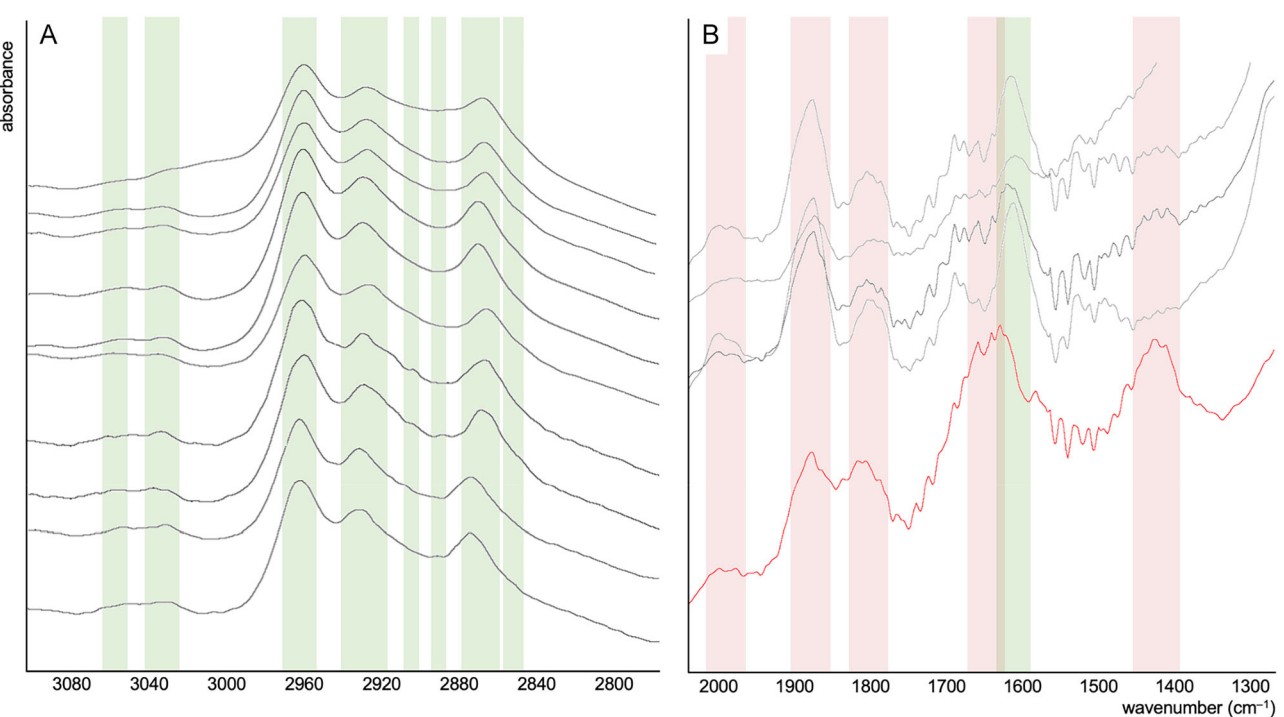

**Fig. 3 | Fourier transform infrared spectroscopy (FTIR) spectroscopy characterisation of Mosquito Creek Formation microbial mats. A** 3100–2800 cm⁻¹ spectral region. **B** 2000–1300 cm⁻¹ spectral region. Green bands denote organic moieties; red bands denote inorganic (matrix silica) contributions. CH₃/CH₂ ratios reported in the text were computed as the relative intensities of the bands at ~2960 cm⁻¹ and 2925 cm⁻¹.

~2890 cm⁻¹, ~2900 cm⁻¹, ~2930 cm⁻¹, ~2963 cm⁻¹, ~3030 cm⁻¹ and ~3055 cm⁻¹ (Fig. 3A) are intense in mat laminations relative to the matrix. Aromatic and aliphatic spectral signatures are virtually absent from the matrix (see Fig. 3B); rare detections within the matrix can be ascribed to fragments of kerogen and other disseminated organic materials derived from the erosion of relatively organic-rich layers. The CH₃/CH₂ ratio (computed as the ratio of intensities of the bands at 2960 cm⁻¹ and 2925 cm⁻¹; cf[30].) of organic layers was found to vary between approximately 0.1 and 1.1, mostly concentrated around 0.4–0.5.

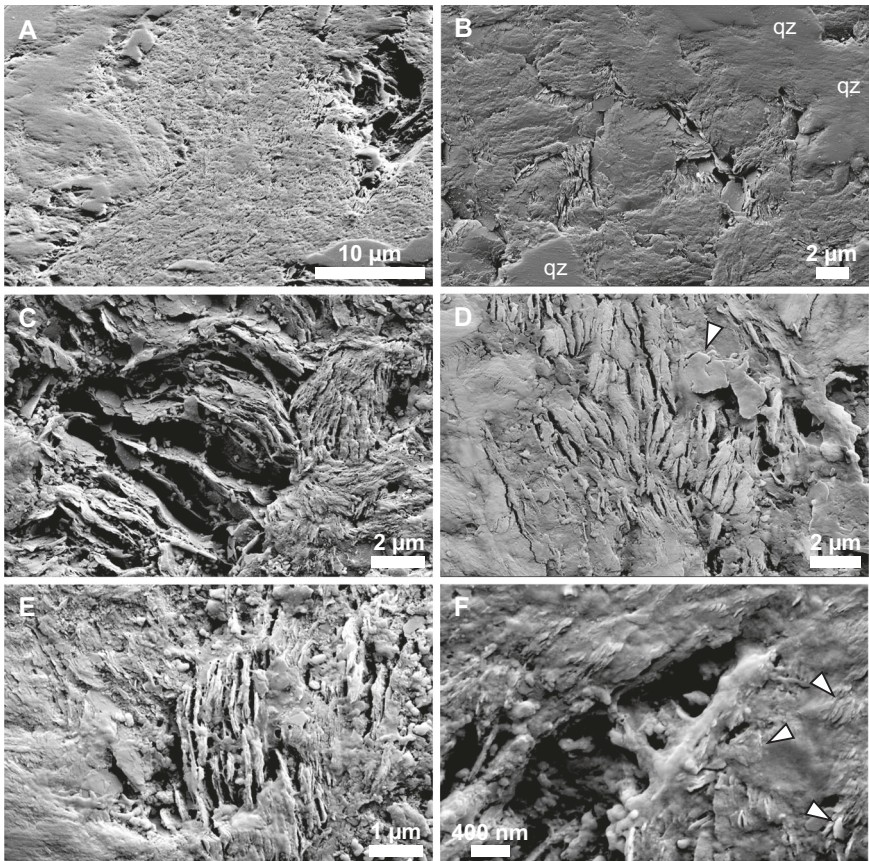

**Fig. 4 | Scanning electron microscopy (SEM) micrographs showing the mineralogical context of Mosquito Creek Formation microbial mats. A** SEM micrograph (image acquired at an angle of 52˚) showing undulatory microbial mat–phyllosilicate layer within an arkosic sandstone matrix. Clay particles are oriented approximately parallel to the microbial mat layer. **B** Phyllosilicate-rich interstitial zones within a quartz (qz) matrix. Microbial mat trends approximately vertically. **C** Oriented domains of illite with highly variable particle size interpreted as the product of illitisation of feldspar or other K-bearing phases. **D** Subaligned phyllosilicate platelets exhibiting quasi-hexagonal form (example arrowed), potentially representing the original morphology of primary kaolinite. The small booklet and subhexagonal morphologies of these minerals suggests kaolinite. **E** High-resolution SEM micrograph showing authigenic nano–micrometric phyllosilicate platelets associated with microbial mat laminations. Microbial mat trends approximately vertically. **F** High-resolution SEM micrograph showing nanometric phyllosilicate platelets (arrowed) embedded within an amorphous carbon-rich material interpreted as extracellular polymeric substances (EPS).

EELS spectral analyses of ultrathin (~60 nm) FIB-milled sections (Figs. 5–6, S27–S30) elucidates the carbonaceous composition and bonding structure of the organic materials at very high resolutions, i.e., with a spot size of 3 nm. C K-edge spectra were found to be typical of turbostratic carbon, characterised by two peaks, one at ~285 eV denoting transitions to the π* molecular orbital, and another at ~290 eV induced by transitions to σ* orbitals (Fig. 64D). EELS mapping of fine-grained, elongate, oriented mineral particles shows that these phases have a composition rich in K, Al, Si and O, i.e. illite-like phyllosilicates (Figs. 5D, S27–30). Small particles of denser mineral phases, including iron sulphides, titanium oxides and rare earth element phosphates, are entrained within laminations (Figs. 5, S29).

High-resolution TEM (HRTEM) micrographs show intimate nanoscopic intercalation between organic materials and subaligned phyllosilicates (Figs. 6A–C, S31–S33). Carbonaceous materials exhibit an amorphous to poorly graphitised ultrastructure rich in polycyclic aromatic hydrocarbons and occur in 10–150 nm layers and domains sandwiched between phyllosilicates, which are generally characterised by an interlayer d-distance of ~0.98–1.05 nm with continuous defect-free fringes (Fig. S33), consistent with the high crystal order of illite as detected using micro-XRD (Figs. S15–S18).

Synchrotron-based STXM was used to elucidate the local electronic structure of carbon (C K-edge at 280–295 eV; Figs. 7A, S34–S35) and nitrogen (N K-edge at 395–410 eV; Fig. 7B) atoms and map these heterogeneities with a spatial resolution of 30 nm (Fig. S36).

Absorption features in C-XANES spectra include: ubiquitous and very prominent peaks at 285.1–285.5 eV assigned to the 1s–π* of aromatic/olefinic carbon (C=C); a rare and weak absorption feature at 285.9 eV interpreted as aromatic phenol or ketone C=O; a rare and weak absorption feature at 286.0 eV potentially reflecting unsaturated aromatic C or C–N; a feature at 286.7 eV ascribed to imine (C=N), nitrile (C≡N), carbonyl (C=O) and/or phenolic (Ar–OH) groups); a common feature at 288.1 eV (occurring as a shoulder on the more intense 288.5 eV peak) consistent with 1s-3π/σ* transition of aliphatic carbon; a ubiquitous and intense feature at 288.5 eV assigned to the 1s– π* carboxyl bond (O–C=O); a rare and weak feature at 290.3 eV assigned to the 1s–π∗ transition of carbonate $CO_3$; and ubiquitous and very prominent peaks at 290.8, 291.4 and 291.8 eV assigned to aromatic groups. N-XANES spectra show a clear N K-edge (i.e., the presence of N) but few to no distinct absorption bands related to specific N-bearing functional groups. Potential signals include: rare and weak features at 398.8 eV potentially reflecting imine, nitrile and/or aromatic groups; and somewhat distinct features at 401.3 eV consistent with amide groups. Although weak, the presence of an unambiguous imine/nitrile peak in the C-XANES spectra suggests that the 401.3 eV peak is real; however, it should be noted that features at the N K-edge are barely distinct from spectral noise. STXM spectra of the matrix mineral phases are dominated by features at 297.5 and 299.9 eV assigned to the $L_{II}$ and $L_{III}$ absorption edges of the K L-edge (Fig. S34); C- and N-bearing compounds are absent from the matrix (Fig. S34).

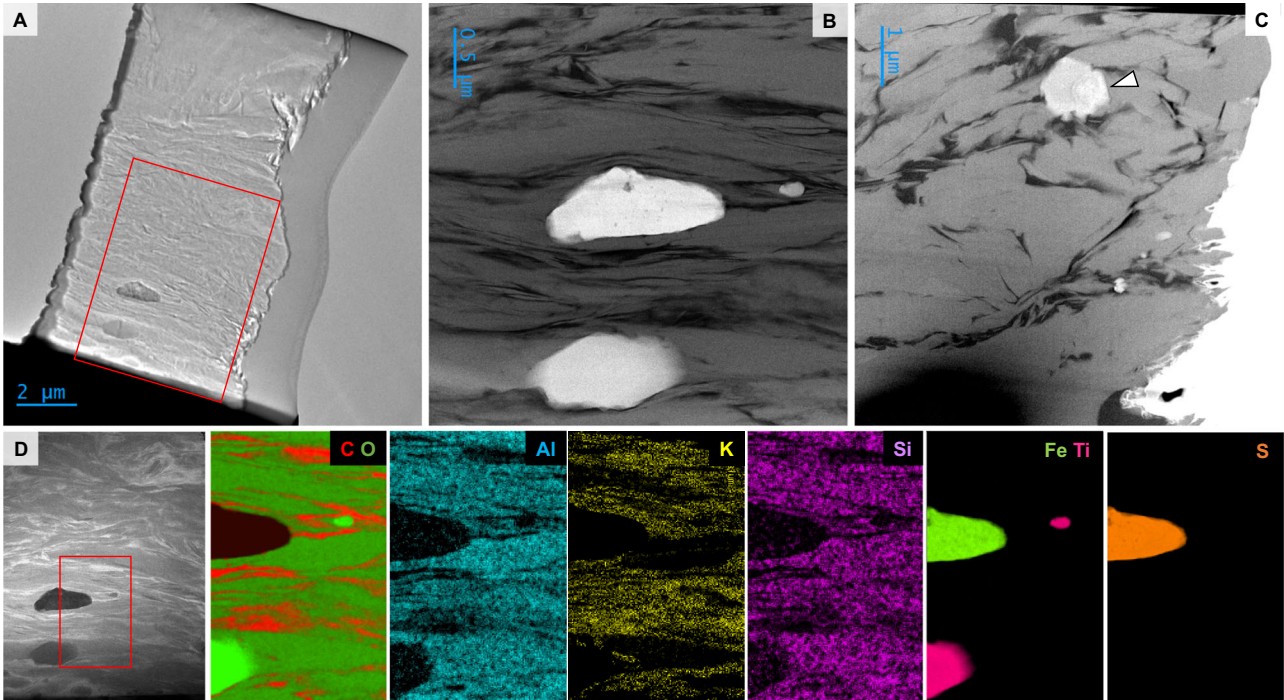

**Fig. 5 | Transmission electron microscopy (TEM) and electron energy loss spectroscopy (EELS) characterisation of Mosquito Creek Formation microbial mats. A** TEM image of an ultrathin (<80 nm) FIB-milled section (sample 28c). Red box shows region mapped in **D**. Scanning TEM (STEM) brightfield images showing carbonaceous materials (black) intercalated between elongate phyllosilicate particles (grey) (**B**, sample 28c; **C**, sample 21b). Panel **B** shows the microstratigraphy of the mat and the manner in which mat laminations trap heavy mineral particles (white). Arrow in **C** denotes a rare earth element phosphate particle (see Fig. S29). **D** STEM darkfield micrograph and EELS compositional maps (sample 28c). Red box in micrograph denotes the region mapped.

NanoSIMS ion mapping shows that carbon (mapped as $^{12}C$ and $^{13}C$) and nitrogen (mapped as $^{12}C^{14}N$) are strongly, but not exactly, correlated within 2–10 μm-thick undulatory organic laminations and absent from the matrix (Figs. 7C, S37). Nitrogen is present at distinctly lower concentrations than carbon. $\delta^{13}C$ ranges between −35.6‰ and −63.4‰.

## Discussion

This study provides a detailed systematic appraisal of an Archaean siliciclastic microbial mat ecosystem from thin section to nanostructural scales. As in other similar siliciclastic sequences on which bulk geochemical analyses have been conducted[10–13], we find evidence for exceptional preservation of primary organic materials. The combination of Raman and FTIR microspectroscopic analyses, HRTEM imaging, and EELS, STXM and NanoSIMS nanoscale geochemical analyses indicate that the MCF microbial mats are preserved as kerogen with amorphous (i.e. without long-range crystalline structure) to poorly ordered, partially graphitised turbostratic (i.e. partially ordered regions of graphene clusters with weak long-range crystalline structure) nanodomains retaining an unprecedented level of primary biogeochemical heterogeneity.

All organic moieties identified have potential and plausible origins within microbial mat-building biomass. Aliphatic organic materials detected using both FTIR and STXM likely originated within the cell wall lipids of mat-building organisms; aliphatic C–H moieties are integral to the longer, relatively unbranched fatty acid-like membranes of bacteria and the shorter, relatively highly branched isoprenoid-based membranes of Bacteria and Archaea[31,32] and have been identified in several other ancient microbial mat horizons[33]. The range of measured $CH_3/CH_2$ values within MCF microbial mats (0.1–1.1, largely concentrated around 0.4–0.5) is most consistent with an origin in long, relatively unbranched membrane lipids (*cf*[30].), suggesting that the palaeoecology of the MCF mats was dominantly bacterial, with minor archaeal components characterised by shorter, relatively highly branched, isoprenoid-like lipids (higher $CH_3/CH_2$ ratios above -0.5). Carbonyl, olefins, and phenols and other aromatic groups are widely biosynthesised and utilised by bacteria[34,35], but some aromatic groups may also represent the product of aromatisation of kerogen during diagenesis and metamorphism[36,37]. Carboxyl groups, in addition to their role in cell envelopes, are ubiquitous in extracellular polymeric substances (EPS), which also contain aromatic, aliphatic and numerous other organic groups[38]. EPS is a dominant constituent of microbial mats[39] and highly resilient to advanced diagenesis and metamorphism[40]; it is therefore anticipated that a portion of the biogeochemical complexity in fossilised microbial mats arises from preserved EPS, potentially some fraction of the aromatic, aliphatic and carboxyl groups identified in FTIR and STXM data. Finally, C–N, C=N and C≡N are essential to nitrogen biogeochemistry and biosynthesis. Similarly to interpretations of isotopic fractionations in disseminated organic materials in shales of the MCF[21], coupled carbon- and nitrogen-cycling metabolisms were likely present in these shallower water ecosystems; eroded microbial biomass from coastal sandstone environments, such as described herein, may have been deposited and concentrated within the deeper water shales described by Koehler et al[21]. In summary, the wide diversity of molecular complexity preserved within MCF microbial mat laminations can be ascribed to distinct parts of the mat-building community, including both cellular and extracellular materials. This richness of biogeochemical complexity and its excellence of retention is consistent with exceptional preservation of microbial mat biomass and is equivalent in diversity and fidelity of organic preservation to almost any known Precambrian shale, phosphate or chert.

This degree of preservation of primary biogeochemistry is both atypical of coarse-grained siliciclastic sedimentary rocks[14] and beyond the expectations of biogeochemical retention in a rock

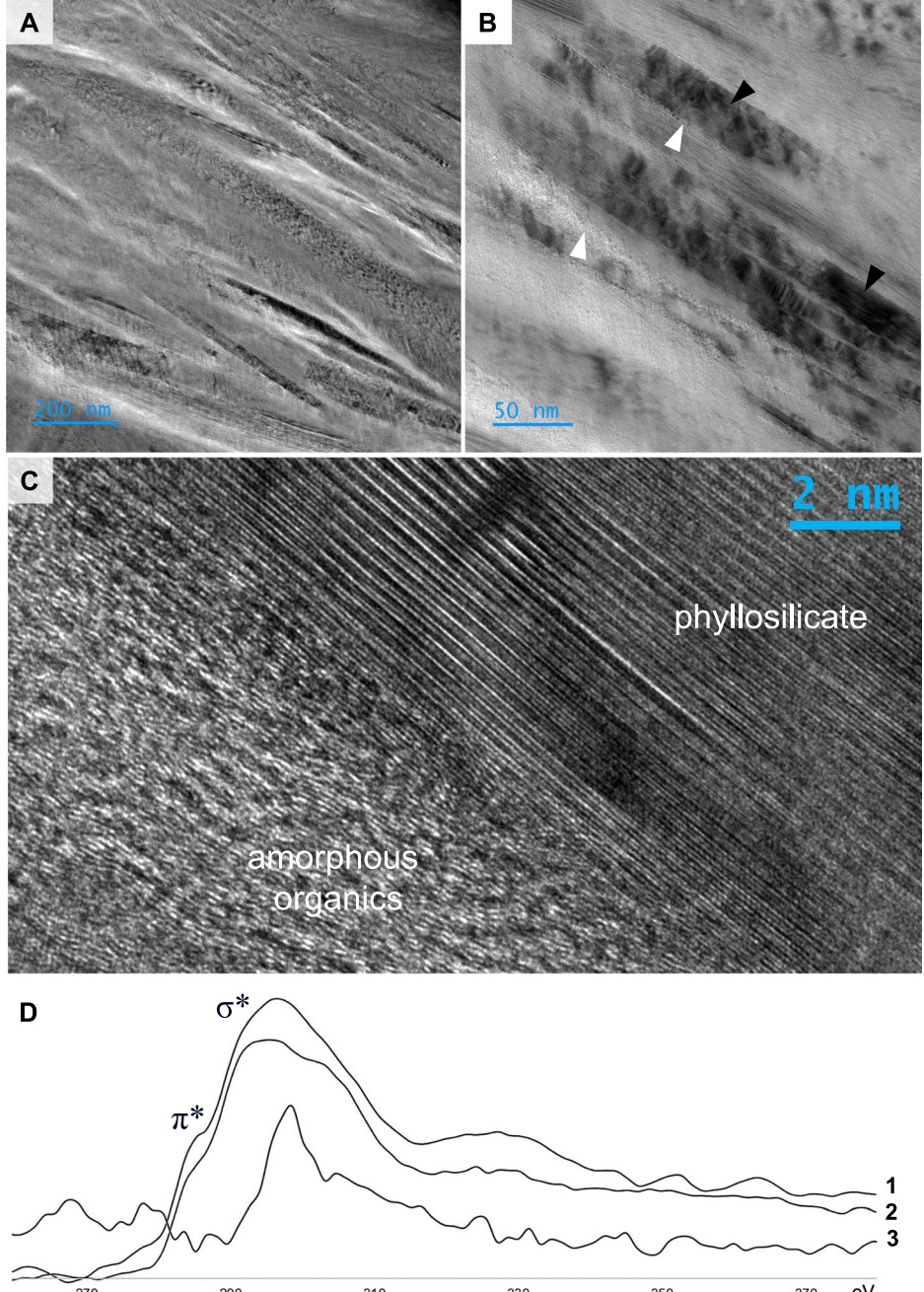

**Fig. 6 | High-resolution characterisation of nanoscale organic preservational phenomena in Mosquito Creek Formation microbial mats. A–C** High-resolution TEM (HRTEM) images showing turbostratic organic materials preserved between illite (phyllosilicate particles). In (B), white arrows indicate organic materials and black arrows indicate illite particles (~1 nm interlayer spacing). **D** Electron energy loss spectroscopy (EELS) spectra obtained from organic materials (1–2, C EELS spectra) and the phyllosilicate matrix (3, K EELS spectrum); spot size 3 nm.

metamorphosed to greenschist facies (cf [41].), suggesting that hitherto unconstrained lithological micromechanics play a role in organic preservation within coarse-grained siliciclastics.

We propose that the remarkable preservation of poorly graphitised turbostratic carbon exhibiting primary biogeochemical heterogeneity observed in the MCF occurs by virtue of depositional and paragenetic processes inherent to arkosic sandstones, with a particular control exerted by burial diagenesis phenomena, specifically the local retardation of carbonisation and graphitisation (i.e. maturation) of organic materials due to volume compensation associated with the illitisation of primary kaolinite [42,43]. The following provides a plausible model (Fig. 8) for the deposition and paragenesis of relatively organic-rich layers within the MCF sandstones:

## Stage 1: sedimentation

The stratigraphy of the upper MCF features continued deposition of thickly bedded (centimetre-scale), coarse-grained, poorly sorted sandstones compositionally dominated by quartz and albite deposited under hydrodynamically energetic shallow marine conditions [15,19]. Finer particles were generally not deposited, but carried further afield in suspension. Periodic low-energy (quiescent) shallow marine episodes resulted in thinner and finer-grained layers dominated by kaolinite, microcline, andesine, pyrite, anatase, rutile, hornblende, chalcopyrite, sodalite and monazite, together with large allochthonous phyllosilicate particles (muscovite, chlorite) that would otherwise be transported further than blocky particles due to their platy morphologies.

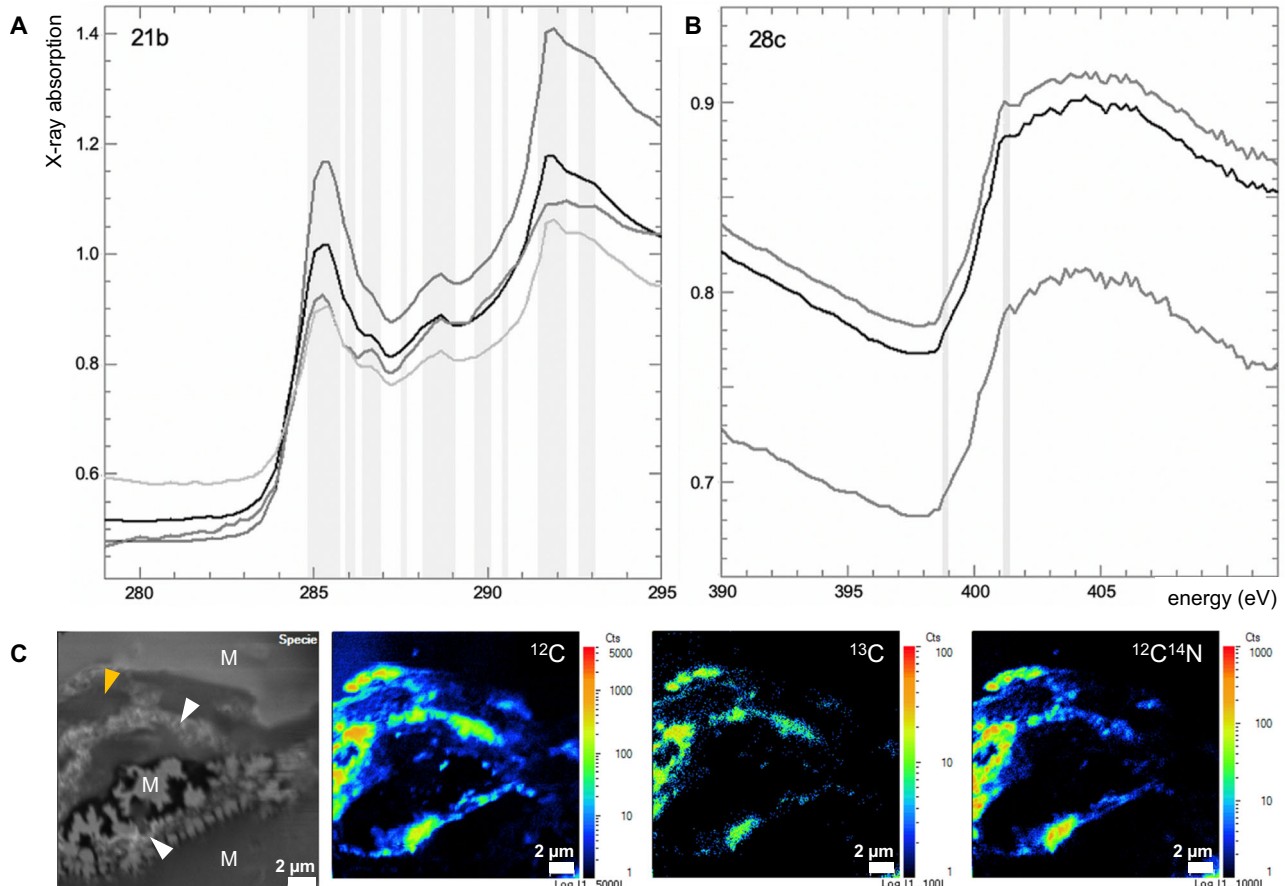

**Fig. 7 | Preservation of biogeochemical complexity in Mosquito Creek microbial mats.** Scanning transmission X-ray microscopy (STXM) analyses of organic materials in FIB-milled ultrathin sections. Absorption spectra correspond to the C *K*-edge (**A**, sample 21b) and N *K*-edge (**B**, sample 28c); spot size 30 nm. See main text for identification of absorption features. **C** Nanoscale secondary ion mass spectrometry (NanoSIMS) analyses showing the correlation of C and N within microbial mat laminations. Optical image (left) shows microbial mat laminations (white arrows) within a phyllosilicate-dominated matrix (regions indicated as M); amber arrow indicates an entrained mineral particle.

## Stage 2: biofilm formation and growth

During quiescent periods of low deposition, microbial biofilms formed together with the deposition of phyllosilicate-rich sediments. Phototrophic biofilms would have successively recolonised successive sedimentary layers to outpace deposition[9], forming the multilayered laminations observed in all samples. Mat–sediment interactions and microbially induced sedimentary structures (e.g. entrapment of silt-sized grains, bifurcation and anastomosis of organic laminae) are testament to simultaneous mat growth and deposition of fine-grained sediment particles. During these low-energy periods, pyrite and chalcopyrite, observed abundantly throughout many samples, may have formed as a result of microbial sulphate reduction and/or other biochemical pathways in the ecosystem.

## Stage 3: development of isolated anoxic microenvironments

A new event (or several events) of high-energy deposition buried the biofilm. Vertical gas and fluid diffusion may have been largely obstructed by new layers of fine deposition and biofilms, such that underlying biofilms, whether still inhabited by non-photosynthetic consortia or already decayed, remained under isolated anoxic conditions that facilitated the exceptional preservation of residual organic materials. These fine-grained layers would not have exhibited the relatively open diagenetic behaviour and diffusion of oxidative fluids that typically flushes out organic materials in coarse-grained sandstones[14].

## Stage 4: burial diagenesis and stabilisation of organic materials

With further burial, the bulk, relatively organic-poor, matrix underwent compaction, calcite cementation and, later, dolomitisation. Poorly developed pressure solution fronts (Fig. 1) suggest minimal compactive stresses during burial diagenesis. In contrast, in the relatively organic-rich layers, weathering of muscovite into illite may have opened up large muscovite particles, forming larger but thinner plates that are increasingly obtrusive to fluid and gas migration. The phyllosilicate-rich fabric associated with biofilm layers would have promoted increasingly anoxic conditions favouring the long-term micro-scale isolation and stabilisation of organic compounds against external degradative influences.

## Stage 5: metamorphism and retention of biogeochemistry

Metamorphism through zeolite to greenschist grade caused the transformation, partial or complete, of K-feldspar grains in the bulk matrix into illite[44], which remained in the interstices of grains and further decreased the porosity of the sandstones. This phase transformation also increased the plasticity of the sediment, enabling it to absorb compactive pressure. Within the biofilms, primary kaolinite underwent transformation into illite. Due to the low connected porosity in the phyllosilicate-rich layers, the illitsation of kaolinite occurred by transformation rather than dissolution and precipitation. For this reason, some minor kaolinite has survived and small, quasi-hexagonal illite plates were observed (reflecting the original morphology of kaolinite). Although minor kaolinite is detected by XRD, no

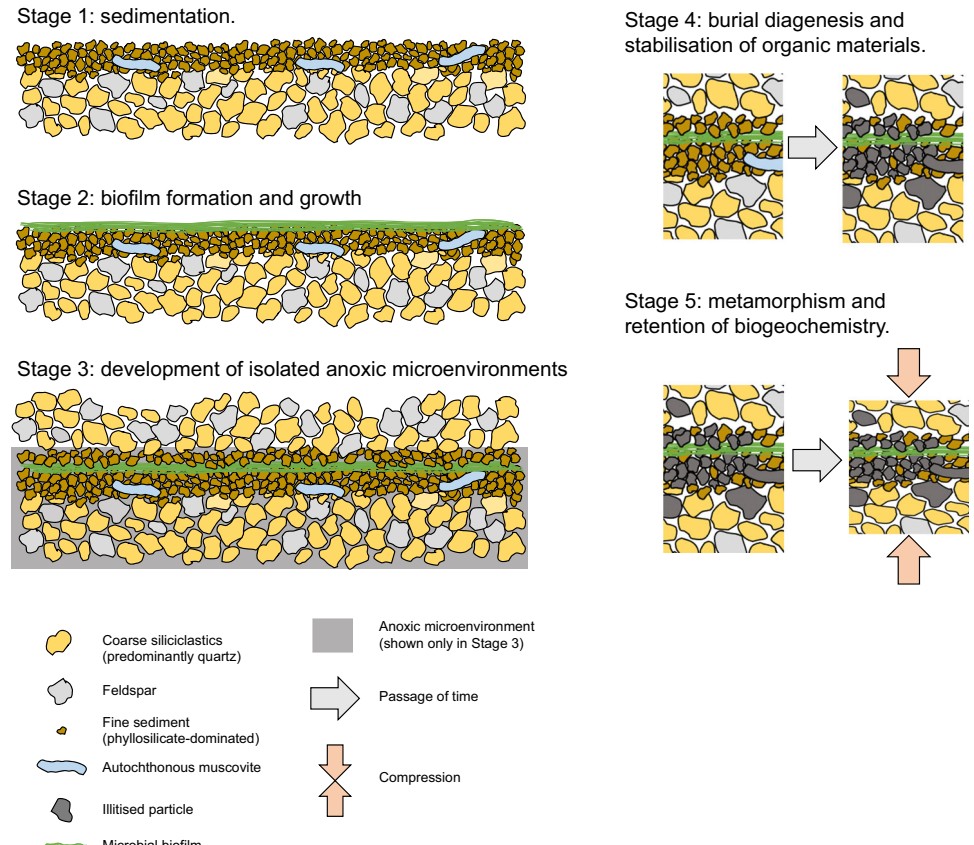

**Fig. 8 | Mechanics of organic preservation in siliciclastic-hosted microbial mats.** Palaeoenvironmental and paragenetic model for the formation of microbial mats and preservation of primary biogeochemical complexity within intercalated organic–phyllosilicate domains of siliciclastic sequences. See main text for details.

pure kaolinite is observed with EDS, which indicates either the intimate mixture of illite and kaolinite plates or incomplete transformation of kaolinite plates into illite. Combined, the volume reduction associated with the bulk matrix and particularly the microbial mat-bearing horizons promoted an environment of reduced pressure (i.e. a layer-specific pressure shadow) that precluded the extensive maturation of microbial mat organic materials and permitted the retention of biogeochemical heterogeneity in amorphous and poorly ordered turbostratic carbon despite regional greenschist grade metamorphism. High-resolution EELS mapping (spatial resolution = 3 nm) and HRTEM imaging (Figs. 5–6) suggests that near-primary amorphous materials occur within nanodomains of preservation intercalated between phyllosilicate particles; slightly lower-resolution STXM mapping (spatial resolution = 30 nm) provides evidence for moderate ordering (turbostratic carbon) surrounding amorphous nanodomains.

Experimental investigations[45,46] have shown that the illitisation of kaolinite occurs progressively under pressure–temperature conditions (0–3000 bars, <300 °C) similar to the lowermost greenschist-grade metamorphism that has influenced the MCF (Stages 4–5 above). Illitisation may cause limited change in the morphologies of phyllosilicate platelets, including a reduction in size[46] and a shift from filamentous and platelet morphologies during earliest diagenesis to lath-like shapes after burial diagenesis[43]; platelets and larger platy phyllosilicate grains are observed in the studied samples (Figs. 4–6). In local regions of the MCF, the formation of pore-filling (bridging) illite initiated at relatively low pressures and temperatures (<500 bars and 250–300 °C[45,46]) may also have contributed to causing the volume of the phyllosilicate fraction of the MCF sandstones to reduce at a sufficient rate so as to retard the maturation of microbial mat organic carbon through compaction (Stage 5). This critical transition interval has been estimated to occur at 120–140 °C[43], whereupon the

thermodynamic destabilisation of alkali feldspars such as microcline could also have contributed to the liberation of K into authigenic phyllosilicates in the MCF sandstones (Stage 5).

The formation of authigenic diagenetic crystalline bridging illite is also associated with a strong reduction in the porosity and permeability of sandstones undergoing burial diagenesis[42,44] and may play a role in preventing the ample diffusion of oxidative fluids that typically flushes out any organic materials preserved within sandstones[14]. Combined, the effects of sediment volume reduction, interstitial pressure compensation and reduction in porosity and permeability resulting from illitisation of precursor phases in the MCF provides a plausible microscale paragenetic process by which poorly graphitised organic materials bearing primary biogeochemical complexity can be retained, relatively unaltered, within moderately metamorphosed sandstones.

Combining Raman geothermometry with the above petrological considerations also constrains the paragenetic process. Carbonaceous materials-based Raman geothermometry indicates peak metamorphic temperatures of 330–350 °C (Fig. S23). This value is consistent with the presence of mixed kaolinite–illite phyllosilicate domains observed in the MCF sandstones, since complete illitisation of kaolinite under typical metamorphic regimes requires higher temperatures of >300 °C and pressures of around 1500 bars[43,46,47]. Such temperatures and pressures may not have been reached in the nanoscopic zones of less graphitised carbon within the microbial mat bearing-horizons (pressure shadows), although the overall sequence and bulk samples may have undergone such metamorphism[15,20]. This further sustains our hypothesis that nano–microscopic zones of exceptional primary organic preservation have endured despite regional metamorphism and are preserved by virtue of local lithological micromechanics.

Whether or not biology (i.e. microbial mats) played some role in the illitisation process (*cf.* 48) is difficult to establish due to equivocal

evidence: the reducing environment rich in biomass required for biological illitisation of kaolinite was certainly present in the MCF sandstones at the time of their deposition and early diagenesis; however, the peak temperatures reached by the formation during burial and metamorphism (~330–350 °C) are also consistent with those required for the abiotic formation of illite[48].

Following deposition, the observed associations between phyllosilicates and well-preserved organic materials likely began as low-permeability organic–clay layers[49] that evolved as organic–clay nanocomposites[50] during burial. Continued overburden and low-grade metamorphism would have led to the interstratification of organic materials between phyllosilicate platelets, as observed in our SEM, STEM–EELS and HRTEM datasets, which bears similarity to subparallel associations of kerogen and illite observed in subgreenschist Triassic–Jurassic shales by Ahn et al.[51]. The gradation observed between illite and organic materials in MCF samples (Figs. 6C, S32) is indicative of the intimate three-dimensional association between these inorganic and organic phases and suggests that some organic materials may have entered within the layered illite structure[51]. This may have further enhanced the atomic and molecular-level preservation of organic materials via isolation from external degradative agents.

In summary, the exceptional preservation of organic materials in the MCF sandstones was primarily promoted by three environmental and diagenetic phenomena:

1. Sequences of slow, low-energy sedimentation interspersed with rapid high-energy sedimentation that allowed a series of biofilm-bearing fine siliciclastic sediments to become protected from oxidising fluids at the sediment–water interface.

2. The increased plasticity of sediment due to progressive illitisation during diagenesis and metamorphism that reduced pressures on fine microbial mat-bearing sediments, affording protection against compaction and rupture during high-energy depositional events and later during burial deformation of the sediment.

3. Illitisation of feldspar that caused a decrease in porosity during burial and increased plasticity in the bulk sandstone, affording protection against metamorphic pressure (volume compensation reducing the compaction of organic materials) and promoted an anoxic microenvironment within fine-grained, microbial mat-bearing layers.

Establishing the lithological mechanisms by which biogeochemical heterogeneities may become preserved in non-traditional biosignature repositories, such as coarse-grained siliciclastics, opens up new avenues for searching for traces of life on the early Earth. The maturation of organic materials in geological sequences is influenced by a number of factors, including temperature, pressure, the passage of oxidising and metamorphic fluids, the inherent graphitisability of the original organic materials, and the lithology of the host rocks[50–53]. Reducing the unfavourable influence of any of these factors over geological timescales may enhance the preservation of organic materials even in very ancient rocks. Structural and chemical characterisation of the MCF sandstones implies that phyllosilicate-rich regions within have minimised the degradative influence of multiple such factors, certainly temperature, pressure and the passage of oxidising fluids. This has a significant implication for biosignature research in deep time, suggesting that geological sequences that have previously been overlooked as archives of biospheric evolution may be revisited at a different scale of observation; namely, the nanoscale as demonstrated herein. This sustains an inference by Brasier et al.[54] that efforts to comprehensively constrain the nature and distribution of the early biosphere should consider non-traditional taphonomic windows, often those in which traces of life are not anticipated to be preserved. Older siliciclastic sequences known to preserve similar microbial mat fossils, such as the 3.22 Ga Moodies Group[10] and portions of the 3.47 Ga Middle Marker horizon[24] should be studied at higher resolution to constrain their microscale taphonomic mechanics with the objective of devising generalised paragenetic and taphonomic models for organic preservation in coarse-grained sedimentary rocks.

Well-preserved associations of phyllosilicates and organic materials have been recognised for decades and direct observations of such nanocomposites in modern and recent sedimentary environments have been extensively made[44,49,50]; however, the micromechanics and nanomechanics of long-term organic preservation in such associations has largely remained subject to speculation. Our study provides evidence for a mechanism by which the preservation of organic materials and their relationships with fabric and mineralogical changes and sediment diagenesis during deep burial on long geological timescales may be achieved. Future research should consider other phyllosilicate-rich environments, particularly those with distinct mineralogies but similar thermal maturities and metamorphic histories, to evaluate the potential universality of the mechanism proposed over billion-year geological timescales. Such studies may unveil further novel taphonomic windows, thereby extending the known range of biosignature repositories in Earth's most ancient rocks.

Understanding modes of preservation in coarse-grained siliciclastics is also highly relevant for the search for life on Mars, since similar sand-grade sedimentary rocks containing organic materials and diverse phyllosilicates are abundant within the Noachian sequences of Mars' equatorial regions and representative examples have recently been collected at the western delta of Jezero crater by the *Perseverance* rover[55,56]. If biogenic organic materials are preserved within the strata of the Jezero delta, they are anticipated to be present in quantities below 0.1 wt%[55] but might preserve similar biogeochemical detail to those of the MCF sandstones if witnessing comparable diagenetic and/or burial conditions. Following Mars Sample Return, a similar level of correlative micro–nanoscale analytical stringency to that described herein will be required to constrain the origins and evolution of such organic materials. The multi-technique, minimally destructive approach developed herein therefore provides a promising blueprint for the analysis of such materials.

## Methods

### Optical microscopy
Optical microscopy was performed using an Olympus BX63 system equipped with ×2, ×5, ×10, ×20 and ×40 objectives and a CCD camera (Imaging and Analysis Centre, NHM London, UK). Images were acquired using Olympus CellSens software.

### Micro-XRD
A Rigaku D/max Rapid2 micro-diffraction system (NHM London) was used to identify crystalline phases in the samples. This system is equipped with a Cu X-ray source, an incident beam monochromator, a pinhole collimator, a manual x–y stage on a two-axis goniometer, and a 2D curved image plate detector. The X-ray tube was operated at 40 kV and 36 mA and the X-ray beam was collimated to provide a 100 μm beam spot.

Polished thin sections were analysed in reflection mode at a constant angle of 5° between the X-ray beam and the sample surface, with diffraction intensities collected between 5° and 70° 2θ. In most measurements, a limited oscillation of the sample of 4–10° was applied perpendicular to the sample surface to increase the number of crystallites exposed to the X-ray beam and to increase the randomness of crystallite orientations. The analysis duration for each sample was 15 minutes.

2D diffraction images were converted to conventional 1-D XRD patterns using the 2DP software (Rigaku). Phase identification was conducted using the Highscore software (Malvern Panalytical) in combination with the ICDD PDF-4 database.

## SEM-EDX

SEM-EDX analyses of large regions (up to several square millimetres) of selected thin sections were performed using a JEOL IT500 system equipped with an Oxford Instruments X-Max EDX detector, operating at an accelerating voltage of 20 kV (Imaging and Analysis Centre, NHM London). Thin sections were uncoated and observed under low-pressure conditions.

High-resolution SEM-EDX analyses of clay minerals in thin section were performed using a Zeiss Ultra Plus field-emission SEM operating at 20 kV and a chamber pressure of 100 Pa equipped with an Oxford Instruments Ultim Max 170 EDX detector enabling rapid and high-sensitivity low-voltage SEM-EDX data acquisition at sub-micrometre spatial resolutions (Imaging and Analysis Centre, NHM London). Two sets of equivalent samples were coated i) with gold for high-resolution imaging and ii) with carbon for EDX measurements.

## Raman microspectroscopy

Raman maps and spectra were acquired using a WITec Alpha 300 R system at the Dipartimento BiGeA, Università di Bologna, Italy, equipped with a 532 nm laser, which was operated using a laser power of 1 mW to avoid the potential thermal alteration of carbonaceous materials. 2D Raman spectroscopic maps were acquired and representative spectra of carbonaceous materials were extracted from these maps for spectral analysis and geothermometry calculations, following the methods described in Kouketsu et al. (2014) (Eq. 1) and Beyssac et al. (2002) (Eq. 2):

(1)  $T(°C) = -2.15(FWHM\text{-}D1) + 478$

(2)  $T(°C) = -445(R2) + 641$

where FWHM-D1 is the full width at half maximum of the D1 peak centred at ~1350 $cm^{-1}$.

## FTIR microspectroscopy

FTIR spectral maps were obtained using a Nicolet iN10 mx FTIR microscope (ThermoFisher Scientific) at the Imaging and Analysis Centre, NHM London. Operating in transmission and using an aperture of 50 μm × 50 μm, maps were acquired with a grid spacing of 50 μm in both axes. Spectra were collected from 4000–675 $cm^{-1}$ with a resolution of 4 $cm^{-1}$ using a liquid nitrogen-cooled MCT/A detector and a KBr beamsplitter. Each spectrum was collected for 19.73 seconds (64 scans). A background spectrum with the same parameters was collected through air every 20 minutes. Individual spectra within kerogenous regions were also acquired; to facilitate a better signal to noise ratio, these were collected for 78.92 seconds (256 scans).

## Focussed ion beam (FIB) sample preparation

Eight sub-samples were prepared from thin sections using FIB milling at the Kelvin Nanocharacterisation Centre, University of Glasgow, UK. Ultrathin samples of <80 nm for TEM−STEM−HRTEM−EELS and STXM analyses were prepared via the lift-out method using low $Ga^+$ ion currents and energies at shallow-angle steps during final ion polishing to prevent damage to the surface layer, such as the formation of gallium implantation artefacts, amorphisation of beam-reactive phases, compositional changes, and redeposition of sputtered material (in this case mostly quartz from the rock matrix and platinum from the protective coating applied to the thin section).

## TEM−STEM−HRTEM−EELS

TEM−STEM−HRTEM−EELS data were performed at the Korea Basic Science Institute, Dajeon, Republic of Korea, using a JEOL Monochromated ARM-200F (NEO-ARM, DJ105) operating at 200 kV. A Gatan imaging filter (GIF) Continuum HR-1066 spectrometer was used to collect electron energy core loss spectra. The probe convergence semi-angle was 23.12 mrad and the collection semi-angle was 29.57 mrad. The EELS spectrum image acquisition (SI) was performed using DualEELS. The spectrum

image was acquired at 150 × 100 pixels with a pixel acquisition time of 0.1 s. Chemical elemental maps were generated using Gatan Digital Micrograph (DM) software.

## Scanning Transmission X-ray Microscopy

Scanning Transmission X-ray Microscopy (STXM) is one of the two end stations of the DEMETER (Dual Electron Microscopy and Spectroscopy) beamline at the National Synchrotron Radiation Centre (SOLARIS), Kraków, Poland. STXM measurements combine spectroscopy and microscopy in a single analytical system with a high spatial resolution. Soft X-ray absorption or near edge X-ray absorption contrast provides high-sensitivity differentiation of species that have similar elemental composition but are chemically distinct. The photon flux at the STXM branch of the Demeter beamline was as follows: $8.62 × 10^{11}$ ph/s for an energy of 285 eV; $8.00 × 10^{11}$ ph/s for an energy of 401 eV. Both values were measured at exit slits with an opening of 30×30 μm. The ring current is 292 mA.

STXM imaging was carried out on samples prepared as ultrathin (<80 nm) sections. Measurements were carried out under environmental conditions (i.e. the pressure within the microscope chamber did not exceed $6 × 10^{-2}$ mbar). The spatial resolution of STXM imaging depends on the quality of the Fresnel zone plate (FZP), in this case 30 nm. In order to obtain high-resolution spectra of the carbon and nitrogen $K$-edges, sequences of images were collected as image stacks. A photomultiplier tube (PMT) with a P-43 scintillator was used to detect the transmitted signals, making it possible to count photons of soft X-rays with high efficiency. For the C $K$-edge, spectral images were collected at an energy step of 0.5 eV between 279 eV and 283 eV, at an energy step of 0.1 between 283 eV and 294 eV, and at an energy step of 0.5 eV between 294 eV and 300 eV. For the N $K$-edge, spectral images were collected at an energy step of 0.5 eV between 394 eV and 396 eV, at an energy step of 0.2 between 396 eV and 404 eV, and at an energy step of 1 eV between 404 eV and 413 eV. The energy step was condensed to 0.1/0.2 eV over the energy range in which the different forms of carbon/nitrogen were expected in order to maximise the sensitivity of STXM microscopy in this range. Dwell times of 2 ms and 3 ms were used for C and N, respectively. All analyses were performed twice; similar intensities and distributions of signals denoted that irradiation damage had not occurred. Stack alignment and data analysis were performed using aXis2000 software. The spectra shown are the average of tens to hundreds of spectra collected within distinct nanometric–micrometric organic-rich regions within FIB sections.

## NanoSIMS−carbon, nitrogen

NanoSIMS analyses were conducted on selected circular thin sections (10 mm and 25 mm diameters) using a CAMECA NanoSIMS 50 at the Korea Basic Science Institute, Busan, Republic of Korea. Regions of analysis were selected to include both organic materials and the surrounding matrix and were re-localised using secondary ion imaging. A focused caesium primary ion ($Cs^+$) beam was scanned over the sample surface with an impact energy of 16 keV. The Cs primary ion beam increases the yield of negative secondary ions. Negative secondary ions from the samples were analysed by mass-to-charge ratio (m/z) using a double-focusing magnetic-sector mass spectrometer. Secondary ions of $^{12}C^-$, $^{13}C^-$, and $^{12}C^{14}N^-$ were detected by each electron multiplier and a high-mass resolution technique with mass resolving power of 7,000 was used to precisely analyse carbon isotopes without mass interference. Areas of either 30 × 30 $μm^2$ or 50 × 50 $μm^2$ were pre-sputtered with a $Cs^+$ primary ion beam with an intensity of 100 pA for 40−60 min to remove contaminants (and the 10 nm gold coating layer applied before analysis) from the sample surface before measurement. Following pre-sputtering, the Cs+ primary ion beam was focused into a 100 nm beam with an intensity of 2 pA and rastered over the area of interest. All ion images were measured with 256 × 256 pixels with a dwell time of 5 ms/pixel. A normal incidence electron flood gun with an

intensity of 2 μA was used for charge compensation during pre-sputtering and analysis. All samples were coated with 10 nm of gold to prevent charging. NanoSIMS analyses were performed under ultra-high vacuum (low $10^{-10}$ Torr).

## Data availability
The authors declare that the data supporting the findings of this study are available within the paper and its supplementary information files. The fossil material is stored in the collections of the Oxford University Museum of Natural History. Methodological information and data treatment processes required to reproduce this study in whole or in part are presented in full in the Methods.

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

## Acknowledgements

K.H.L. acknowledges funding from the UK Space Agency (grant nos. ST/V00560X/1 and ST/Z000491/1). Access to KBSI facilities was enabled by Europlanet funding (grant nos. 20-EPN2-108 to K.H.L. and NRF-2020K1A3A1A78089517 to K.Y.) and Korea Basic Science Institute Visiting Programme funding (grant no. C330430) to K.H.L. J.H.J. is supported by funds (grant nos. A412310 and A412550) from the Korea Basic Science Institute. Europlanet 2024 RI has received funding from the European Union's Horizon 2020 research and innovation programme under grant agreement No 871149. This publication was partially developed under the provision of the Polish Ministry and Higher Education project support for research and development with the use of research infrastructure at the National Synchrotron Radiation Centre SOLARIS (contract no. 1/SOL/2021/2) and K.H.L. acknowledges SOLARIS beamtime (proposal no. 221903). Duncan Murdock (OUMNH) and Euan Furness (Imperial College London) kindly facilitated access to samples. We also thank William Smith (University of Glasgow) for producing FIB samples, Innes Clatworthy (NHM) for support during the acquisition of SEM-EDX data, and Tolek Tyliszczak (Lawrence Berkeley National Laboratory) for support during the acquisition of STXM data.

## Author contributions

K.H.L. conceptualised the study, conducted optical microscopy analyses, and wrote the manuscript. K.H.L. and J.C. conducted SEM and SEM-EDX analyses. J.N., K.H.L. and J.C. conducted micro-XRD analyses. K.H.L. and W.M. conducted FTIR analyses. K.H.L. and B.C. conducted Raman spectroscopy analyses. K.Y. prepared samples for NanoSIMS analyses. T.E.H. and M.B. conducted NanoSIMS analyses. J.H.J., M.Y.C. and Y.K.S. conducted STEM–EELS and HRTEM analyses. K.M., B.W. and K.H.L. conducted STXM analyses. K.H.L., J.C., K.Y., T.E.H., M.B., J.H.J., M.Y.C., Y.K.S., J.N., W.M., K.M., B.W., C.L.S. and B.C. contributed to the discussion of data and writing and editing of the manuscript.

## Competing interests

The authors declare no competing interests.

### Ethics

All geological materials were collected by the Archaean Biosphere Drilling Project in a responsible manner and in accordance with relevant permits and local laws. Local and regional research relevant to this study has been cited where appropriate.

## Additional information

[1]School of Natural Sciences, Birkbeck, University of London, Malet StreetBloomsbury London WC1E 7HX, UK. [2]Department of Earth Science and Engineering, Imperial College London, London SW7 2BX, UK. [3]Natural History Museum, Cromwell Road, London SW7 5BD, UK. [4]Korea Basic Science Institute, 162, Yeongudanji-ro, Ochang-eup, Cheongwon-gu, Cheongju-si, Chungcheongbuk-do 28119, Republic of Korea. [5]Korea Basic Science Institute, 60, Gwahaksandan 1-ro, Gangseo-gu, Busan, Republic of Korea. [6]Korea Basic Science Institute, 169-148, Gwahak-ro, Yuseong-gu, Daejeon, Republic of Korea. [7]Department of Energy Science, Sungkyunkwan University, 2066, Seobu-ro, Jangan-gu, Suwon, Republic of Korea. [8]National NanoFab Center, 291 Daehak-ro, Yuseong-gu, Daejeon 34141, Republic of Korea. [9]SOLARIS, National Synchrotron Radiation Centre, Jagiellonian University, Czerwone Maki 98, 30-392 Kraków, Poland. [10]School of Geographical and Earth Sciences, University of Glasgow, Glasgow G12 8QQ, UK. [11]Dipartimento di Scienze Biologiche, Geologiche e Ambientali, Università di Bologna, via Zamboni 67, I-40126 Bologna, Italy. [12]Department of Geology, University of Johannesburg, PO Box 524Auckland Park 2006 Johannesburg, South Africa. ✉e-mail: k.hickman-lewis@bbk.ac.uk

