## [Peer Review file · Nature Communications]

Aluminous phyllosilicates promote exceptional nanoscale preservation of biogeochemical heterogeneities in Archaean siliciclastic microbial mats

Corresponding Author: Dr Keyron Hickman-Lewis

Version 0:

Reviewer comments:

Reviewer #1

(Remarks to the Author)

This manuscript presents the first description of exquisitely-preserved microbial mats in 2.9 billion year old sandstones, including mineralogy, organic molecules, and organic isotopes. The manuscript also presents a model for preserving detailed carbonaceous biosignatures in metamorphosed sandstones. In the search for early traces of life, sandstones such as the samples examined are typically overlooked in favor of chemical sediments, especially when searching for organic material. This study shows that under certain common patterns of sedimentation and diagenesis, many Precambrian clastic deposits can also yield detailed biogeochemical data. The results of this manuscript greatly expand the number of locations available to study organic biosignatures throughout deep time and potentially in astrobiology.

Evidence is presented that the illitization of kaolinite abated post-depositional compression in fine-grained mat layers, preventing the graphitization of organic matter which often destroys detailed biosignatures. The multi-step model is supported by sedimentology, petrography, and high-resolution geochemical analyses that examine nanoscale relationships between phyllosilicate minerals and organic carbon in sandstones. Each step of the model involves multiple lines of evidence: for example, burial temperatures are corroborated by silicate mineralogy and Raman geothermometry of organic phases. The evidence not only supports the paper's conclusions, but lays out criteria that can be applied to many other clastic deposits: intermittent calm and energetic deposition, visible evidence for microbial mats, and the transition from kaolinite to illite (two common phyllosilicate minerals). In short, the results are convincing and help advance the field of Precambrian geobiology.

Dylan Wilmeth

Reviewer #2

(Remarks to the Author)

Reviewer Report for the Manuscript: "Aluminous phyllosilicates promote exceptional nanoscale preservation of biogeochemical heterogeneities in Archaean siliciclastic microbial mats"

Summary: The manuscript provides a comprehensive study on the nanoscale preservation of biogeochemical heterogeneities in Archean microbial mats from the MCF formation. Using an array of advanced techniques, such as Raman spectroscopy, FTIR, HRTEM, STXM, and NanoSIMS, the authors present evidence of exceptional preservation of poorly ordered carbon (kerogen) of microbial mats in coarse-grained siliciclastic rocks. The authors convincingly argue that aluminous phyllosilicates play a crucial role in preserving these biogeochemical signatures, especially illitization of kaolinite during burial diagenesis, even in coarse-grained siliciclastic sediments.

The manuscript is strong in its technical approach and breadth of analyses. While fossilized microbial biomass in coarser-grained siliciclastic rocks has been studied previously, this work provides the first systematic examination spanning from thin sections to nanostructural scales. This thorough methodology effectively captures the molecular complexity of preserved microbial remains, delivering valuable insights into Archaean ecosystems and the fossilization processes of microbial mats under high-grade metamorphic conditions.

The following suggestions highlight areas where more detailed interpretations or explanations could enhance the manuscript. These specific comments are intended to help the authors improve clarity and address potential issues.

Comments:

Sandstones and preservation potential

The manuscript emphasizes that coarse-grained siliciclastic sandstones have an unexplored potential for preserving organic matter. However, the findings suggest that preservation primarily depends on fine-grained aluminous phyllosilicates (e.g., kaolinite and illite), likely formed during low-energy depositional episodes. It's well-documented that clay minerals play a significant role in preserving organic matter by trapping and stabilizing organic molecules via adsorption, physical entrapment, and chemical bonding.

Suggestion: Clarify the novelty of this preservation mechanism. Are the aluminous phyllosilicates and the specific diagenetic processes (e.g., illitization) in the Archean MCF siliciclastics different from the ones occurring in worldwide fine-grained environments? It may be helpful to compare this mechanism with existing literature on clay mineral-driven preservation in siliciclastic settings (e.g., Kennedy et al., 2002; Keil & Mayer, 2014) to illustrate how this mechanism differs or builds on past models.

Preservation Mechanism

The study suggests that authigenic phyllosilicates inhibit the carbonization of organic materials by "pressure compensation" linked to the kaolinite–illite transformation. This is plausible, but other factors, such as the low thermal conductivity of clay-rich layers, may also contribute to preserving organic material by creating localized, cooler microenvironments within phyllosilicate-rich zones. Such conditions could reduce thermal degradation in phyllosilicate-rich layers relative to the more porous, thermally conductive sandstone layers.

Suggestion: Since burial and metamorphism likely affected the entire stratigraphic section (sandstones and phyllosilicates), and the only temperature reading (350°C) was from organic-rich phyllosilicate layers, consider estimating the temperature of the organic-poor microbial mats within sandstones (e.g., Fig. S2). While the organic content in these sandstones is considerably lower, obtaining a single temperature reading from organics that are less shielded by phyllosilicates could allow for a comparison and clarify any potential thermal and maturity disparity.

Suggestion: since phyllosilicates are common across various geological settings, it would be valuable to assess whether the MCF phyllosilicates exhibit specific chemical or physical characteristics that might enhance preservation. Are the MCF phyllosilicates or the illitization processes during diagenesis distinct from those in other formations, or might these mechanisms apply more broadly to siliciclastic environments worldwide?

Minor Comments:

- Line 45, 340, 396, 421: The term 'graphitization' is typically associated with temperatures >650°C. A possibility would be to use the term 'maturation' in some cases (lines mentioned) instead.
- Line 63: consider adding Noffke et al., 2002 and 2006 papers to the 2022 reference, as it is important to mention that fossilized microbial biomass has been described in coarser-grained siliciclastic sedimentary rocks for decades.
- Line 101 and Supplementary Data (Geological setting): The terms "organic-rich" and "organic-poor" microbial mats could benefit from quantification. Is there a Total Organic Carbon (TOC) measurement available?
- Line 202: The use of terms like "poorly ordered, partially graphitised turbostratic carbon" is appropriate, but it would be helpful to include a brief explanation for readers, particularly for those who may not be familiar with the significance of this type of kerogen in relation to microbial preservation who may not be familiar with these specific terms.
- Line 300: The manuscript mentions extracellular polymeric substances (EPS) as the dominant constituent in the study microbial mats. Given the known resilience of EPS to diagenesis, this is a plausible hypothesis. Are there specific molecular markers or features in the spectroscopic data that point explicitly to EPS preservation, rather than just bulk organic material?
- Line 317: Compare $\delta^{13}\text{C}$ fractionation values to known microbial metabolic pathways to contextualize these results.
- Line 365: If primary productivity centers around anoxygenic photosynthesis (indicating anoxic conditions) (line 329), wouldn't stage 3 of the depositional model be implicit?
- Line 714: Equation 1 is from Kouketsu et al. (2014) and Equation 2 is from Beyssac et al. (2002).

Reviewer #3

(Remarks to the Author)

This paper presents a detailed study of organic remnants of ancient life in a coarse-grained siliciclastic sedimentary rock. As the authors point out, such rocks have often been ignored, because it is generally assumed that traces of life are poorly preserved within them. This type of sediment represents an energetic depositional environment where low-density organic materials are not easily retained. Also, during diagenesis these rocks would be easily be penetrated by oxidative fluids. However, some intriguing observations suggest that there are mechanisms for preservation in such rocks.

The current paper presents a very detailed study of exactly such a mechanism. The authors studied drill core samples of a sandstone from the 2.97 Ga Mosquito Creek Formation in Australia. They discovered organic layers associated with aluminous phyllosilicates. First they used an impressive array of high-resolution techniques to structurally, chemically and

isotopically characterize these organic layers. Then they focused on the aluminosilicates themselves, and established that kaolinite-to-illite conversion during diagenesis is one of the key mechanisms for the preservation of the organic materials. This illitization caused volume reduction of the rock matrix, preventing extensive graphitization of the organic fractions. It also prevented fluid circulation through pore spaces.

The authors describe in detail how this mechanism would have led to pockets of well-preserved microbial mats. This study thus greatly expands the possibilities of finding traces of life in yet another type of Archean rocks.

I think this is an important study that should be published. However, there are some issues that require minor to moderate revisions. My comments below can hopefully be used to strengthen the manuscript.

Comments:

1) Interpretation of the organic layers:

The organic layers are described in detail in Lines 131-151. These descriptions are key for the interpretation that these represent biofilms/microbial mats. I think there are some shortcomings here. Some of the descriptions in the text are difficult to see in the figures, and it is not adequately discussed how these observations exclude alternative explanations, particularly the possibility that some of the observed organic layers may represent veins of organic fluids.

Important observations in the text here are the following. First there is the mention that the layers bifurcate/anastomose around sediment grains. (Fig.1B-C). Also, in line 138-139 it is said that the layers deform beneath overlying sediment particles and exhibit local rip-ups, tear-ups and crumpling. And in line 140 it is said that sedimentary particles are oriented parallel to laminae (Fig.1D). I carefully looked at these figures 1A-D, and I have a hard time seeing what is described here. This likely has to do with the small size of the images in Fig.1, and lack of annotations. The figures S2-S6 are more detailed, but it would be good if it can be specifically pointed out where in these images the key observations are made that are described in this part of the text (line 131-151).

Also, I miss zoomed-out images of the actual extend of the layering. Only pictures of the thin sections are shown. Are there images of the cores themselves? It should be possible for a reader to see these important observations in a figure in the main paper itself.

In line 147-150 it is written: "On the basis of their sedimentary context, organic composition and morphological identification to known similar structures from throughout the geological record (refs...) we interpret these laminated structures as fossilized siliciclastic microbial mats.

After this sentence, I miss a bit more discussion on alternative explanations for the observed organic layers. The authors quickly jump to the conclusion that these represent microbial mats, without discussing the possibility that biogenic organic fluids could have migrated layer-parallel through the rock. Have these rocks been influenced by hydrothermal fluids? Can this effectively be excluded that these represent layer-parallel organic veins?

I'm confident that the authors can establish this. I'm mostly asking for more clarity, and possibly some annotations in the figures.

2) Raman spectra of the organic phases:

The use of the Raman-based geothermometers is unclear.

In line 156-159 it is established, based on Raman spectroscopic analysis, that the carbonaceous phases experienced a peak metamorphic temperature of 343-348 C. This is extremely precise, and apparently was based on two geothermometers, Beyssac et al. (2002) and Kouketsu et al. (2014). It would be amazing if these two different geothermometers would reach the same temperature with such a small error margin.

I tried to find the details of the calculations made by the authors, and the only thing I found was a mention of the geothermometers in the Methods section, lines 715-716. In fact, the geothermometer of Beyssac et al. (2002) is incorrect as it is listed there for two reasons: 1) Apparently peak-intensities were used, while peak-areas should have been used, 2) Only the D1 and G-peaks were used, while the D2-peak (shown in the spectrum in Fig.S23) was ignored.

So the authors should list the correct formula (as it is given in Beyssac et al., 2002) for this geothermometer in line 716, and they should show (maybe in the supplement) how exactly the calculation was made for this particular geothermometer.

Then, for the Kouketsu geothermometer, the correct formula is used, but it should be shown what the exact temperature is that is obtained with this particular geothermometer. It would be very nice if indeed this gives the same temperature as the one of Beyssac et al. (2002), but this should be verifiable.

3) FTIR data:

In lines 191-199 the results of FTIR micro spectroscopy are presented. Various important organic bonds and groups are found (aromatics, aliphatics, etc.). However, there is no reference to any figure here. The only thing I found were the FTIR

maps in Fig.1E. It would be good to see the FTIR spectra from which we can actually determine whether aromatic and aliphatic groups are present. Also, in line 1970-199 it is said that based on averaging 30 spectra, the CH₃/CH₂ ratio in the organic layers could be established. How was this done? And again, where are the spectra?

4) NanoSIMS d¹³C analysis:

The carbon isotope analysis by NanoSIMS is poorly described. Nothing is said in the Methods section about the carbon isotope standards that were used, nor of the count rates that would constitute a significant signal above background. Also, what is the error on the analysis? With NanoSIMS this can easily be 5 per mil or more. It is important for a reader to be able to independently evaluate the quality of the data.

The NanoSIMS maps in Figure S37 only show maximum counts of ions. It is very difficult to derive a meaningful d¹³C value from this.

In the main text, in lines 252-253 the d¹³C values of the organic layers are presented. The range of d¹³C= -35 to -63 per mil. This lower limit is extremely low. Can the authors verify that this number is correct?

5) Interpretation of metabolism, based on d¹³C

In the Discussion, lines 317-329, the authors describe a range of metabolisms that could be responsible for the observed d¹³C range. These include anoxygenic photosynthesis, methanogens, methanotrophs, sulphate-reducing pathways, and anaerobic nitrogen fixation. That is quite a list, and again a d¹³C= -63 per mil is really extremely low. In my opinion such a low number would specifically suggest methanotrophy.

However, at the end of this section, in line 328-239, the authors draw the conclusion: " Coupling stratigraphic context, inorganic geochemistry and d¹³C measurements suggests that primary productivity in the mats was likely centered on anoxygenic photosynthesis."

How can this conclusion actually be drawn, based on the available information? I would think it is quite difficult to narrow down such a list of possible metabolisms to a single one. I think this is over-interpretation. In any case, it needs to be better explained why this particular metabolism can be inferred.

Version 1:

Reviewer comments:

Reviewer #2

(Remarks to the Author)

I have reviewed the revised manuscript and the authors' responses to the reviewers' comments. The authors have thoroughly addressed all the concerns and suggestions raised by both myself and the other reviewers. I thank the authors for the significant effort they have put into this work. This manuscript makes a valuable contribution to our understanding of biogeochemical signatures in Archean microbial communities. I am therefore pleased to recommend it for publication.

Fuencisla Cañadas

Reviewer #3

(Remarks to the Author)

The authors have produced a very nice revised manuscript, taking into account all my earlier comments, and the comments of two other reviewers. Below I discuss my earlier points, and the replies by the authors. I have only one suggestion, regarding point 3 below (FTIR data). Other than that I think this manuscript can be accepted for publication without any further adjustments.

My earlier comments:

1) Interpretation of the organic layers:

a) Specific evidence in the images.

Figure 1 is now separated into two figures. The new figure 1 contains annotated microscopy images (from the earlier Supplementary figures). There are now clearly described examples of undulatory laminations, bifurcation and anastomosis, entrainment of sedimentary particles, rip/roll structures and microbial mats on top of sedimentary layers.

Figure 2 now contains the Raman and FTIR maps of the carbonaceous materials.

b) Zoom-out of the actual extent of the layering.

It's a bit disappointing that there are no images of the core samples themselves. This means that the extent of the layering cannot be estimated on a larger scale. However, it is clear from the provided images that indeed there are well-preserved remnants of microbial mats.

c) Alternative explanations for the observed organic layers.

This has now been added in lines 172-188. Three important points are now made. 1) There is no petrographic evidence for the emplacement of pyrobitumen. 2) There is no evidence for carbon-rich sediments linked to hydrothermal fluids and vein systems. 3) There are no REE indicators for hydrothermal processes.

2) Raman spectra of the organic phases:

The issue with the Beyssac et al. (2002) geothermometer has now been solved. Also, new temperatures were calculated, and in the supplementary materials (Fig.S23) it is now shown how the peak-deconvolution was done.

3) FTIR data:

I was wondering why the spectra were not shown. Also, it was not clear how CH₃/CH₂ were calculated. There is now a new figure 3, showing the key regions 3100-2800 cm⁻¹ and 2000-1300 cm⁻¹.

What I still don't understand is why it is not explained in lines 269-271 how the CH₃/CH₂ ratio was calculated. I assume that this was based on the intensities of the bands 2960 cm⁻¹ and 2925 cm⁻¹ respectively. I suggest the authors add this information here in the text. This ratio is described in e.g. Iqisu et al. (2009) *Precambrian Research* 173, 19-26. or in Qu et al. (2015) *Astrobiology* 15, 825-842. Also, in the caption of Fig.3. it could be mentioned which IR bands were used exactly for this calculation.

4 and 5) Interpretation of the NanoSIMS and d¹³C analysis

I agree with the authors that the interpretation of the d¹³C data can be removed. Indeed it detracts from the key point of this paper, namely the exceptional preservation of these organic remnants in relatively coarse-grained sediments.

Reviewer #1

This manuscript presents the first description of exquisitely-preserved microbial mats in 2.9 billion year old sandstones, including mineralogy, organic molecules, and organic isotopes. The manuscript also presents a model for preserving detailed carbonaceous biosignatures in metamorphosed sandstones. In the search for early traces of life, sandstones such as the samples examined are typically overlooked in favor of chemical sediments, especially when searching for organic material. This study shows that under certain common patterns of sedimentation and diagenesis, many Precambrian clastic deposits can also yield detailed biogeochemical data. The results of this manuscript greatly expand the number of locations available to study organic biosignatures throughout deep time and potentially in astrobiology.

Evidence is presented that the illitization of kaolinite abated post-depositional compression in fine-grained mat layers, preventing the graphitization of organic matter which often destroys detailed biosignatures. The multi-step model is supported by sedimentology, petrography, and high-resolution geochemical analyses that examine nanoscale relationships between phyllosilicate minerals and organic carbon in sandstones. Each step of the model involves multiple lines of evidence: for example, burial temperatures are corroborated by silicate mineralogy and Raman geothermometry of organic phases. The evidence not only supports the paper's conclusions, but lays out criteria that can be applied to many other clastic deposits: intermittent calm and energetic deposition, visible evidence for microbial mats, and the transition from kaolinite to illite (two common phyllosilicate minerals). In short, the results are convincing and help advance the field of Precambrian geobiology. I only have a few minor notes, ones that do not diminish the comments above but didn't quite fit with the broad notes.

Dear Dylan, thank you very much for your review of our manuscript and your constructive notes below. We appreciate your positive evaluation of our manuscript, and hope that it might encourage similarly detailed descriptions of nanoscale characteristics in other Precambrian biosignature-bearing horizons.

1: Line 58: I would advise the phrase "Most evidence for life in the Precambrian..." to be changed to "Most microfossils in the Precambrian..." or "Most organic biosignatures in the Precambrian...". While Precambrian cherts, phosphates, and shales do preserve many microfossils, macroscopic fossils such as stromatolites and other microbialites are far more abundant in most Precambrian deposits. Other references to fossils in cherts, shales, etc. in the manuscript are fine.

We agree that this change improves the accuracy of the statement and have edited as recommended to "Most organic biosignatures in the Precambrian..."

2: Line 205: The phrasing of this sentence confused me- I believe the authors meant to write "oriented mineral particles shows..."

Apologies for this unclear phrasing – we have edited as recommended to "...elongate, oriented mineral particles shows...".

One final point that should be taken as a question rather than a criticism: the manuscript lists the range of carbon isotopes analyzed, but not nitrogen isotopes, though they were measured and briefly discussed. If space merits, similar values should be given for d15N. Discussion comparing results to specific 15N values of previous studies (Homann et al., 2018, Stueken & Buick, 2018, Koehler et al., 2019, already cited) could provide even more evidence of metabolic variability, as presented with d13C from Lines 317-329.

Our NanoSIMS analyses mapped the distributions of both carbon and nitrogen. Unfortunately, the low counts of nitrogen obtained were insufficient to calculate a reliable nitrogen isotope

ratio (the errors were very large in all samples studied). We therefore chose to show only the mapping data as an indication of the presence of nitrogen (as confirmed by STXM data) but cannot provide further detail on this point at this time. We suspect that the nitrogen isotope data obtained on other parts of the Mosquito Creek Formation by Koehler et al. (2019) were made possible by the higher concentrations of nitrogen present in the shales studied in that work relative to the siliciclastics studied herein. Sample preparation likely also plays a role – Koehler et al. (2019) studied organic-enriched bulk residues whereas our samples are polished thin sections. The advantage of the former is that greater quantities of (nitrogen-bearing) organic materials can be obtained; the advantage of the latter is that elemental abundances can be tied directly to morphological structures (i.e., microbial mats, which were not noted or characterised by Koehler et al.).

Following comments from the other reviewers, we have elected to remove the carbon isotope fractionations from this manuscript. All reviewers have requested further details on this point, and (also since we are unable to provide nitrogen isotope values at present) answering these comments satisfactorily would require the addition of a separate section to the discussion, which would be secondary to, and perhaps detract from, the main purpose of this manuscript – to describe the nanomechanical processes promoting exceptional organic preservation in these samples. Additionally, since we do not wish this manuscript to become discussion of the utility of carbon isotopes on ancient samples, and since we were only able to acquire a small number of $\delta^{13}\text{C}$ measurements (due to the time-consuming nature of the NanoSIMS analyses conducted), hence the discussion would be inherently incomplete. We are currently working to obtain coupled carbon, nitrogen and sulphur isotope characterisation of a larger number of Mosquito Creek Formation samples for a future manuscript.

We hope that these responses satisfy any remaining concerns. Thank you again for your constructive review of our manuscript.

Reviewer #2

Reviewer Report for the Manuscript: "Aluminous phyllosilicates promote exceptional nanoscale preservation of biogeochemical heterogeneities in Archaean siliciclastic microbial mats"

Summary: The manuscript provides a comprehensive study on the nanoscale preservation of biogeochemical heterogeneities in Archean microbial mats from the MCF formation. Using an array of advanced techniques, such as Raman spectroscopy, FTIR, HRTEM, STXM, and NanoSIMS, the authors present evidence of exceptional preservation of poorly ordered carbon (kerogen) of microbial mats in coarse-grained siliciclastic rocks. The authors convincingly argue that aluminous phyllosilicates play a crucial role in preserving these biogeochemical signatures, especially illitization of kaolinite during burial diagenesis, even in coarse-grained siliciclastic sediments.

The manuscript is strong in its technical approach and breadth of analyses. While fossilized microbial biomass in coarser-grained siliciclastic rocks has been studied previously, this work provides the first systematic examination spanning from thin sections to nanostructural scales. This thorough methodology effectively captures the molecular complexity of preserved microbial remains, delivering valuable insights into Archaean ecosystems and the fossilization processes of microbial mats under high-grade metamorphic conditions.

The following suggestions highlight areas where more detailed interpretations or explanations could enhance the manuscript. These specific comments are intended to help the authors improve clarity and address potential issues.

Thank you very much for your positive evaluation and highly constructive review of our manuscript and the suggestions for improvement and clarification. We hope that the changes and additions made to the revised manuscript address your concerns.

Comments:

Sandstones and preservation potential. The manuscript emphasizes that coarse-grained siliciclastic sandstones have an unexplored potential for preserving organic matter. However, the findings suggest that preservation primarily depends on fine-grained aluminous phyllosilicates (e.g., kaolinite and illite), likely formed during low-energy depositional episodes. It's well-documented that clay minerals play a significant role in preserving organic matter by trapping and stabilizing organic molecules via adsorption, physical entrapment, and chemical bonding.

Suggestion: Clarify the novelty of this preservation mechanism. Are the aluminous phyllosilicates and the specific diagenetic processes (e.g., illitization) in the Archean MCF siliciclastics different from the ones occurring in worldwide fine-grained environments? It may be helpful to compare this mechanism with existing literature on clay mineral-driven preservation in siliciclastic settings (e.g., Kennedy et al., 2002; Keil & Mayer, 2014) to illustrate how this mechanism differs or builds on past models.

Thank you for highlighting the opportunity to expand upon this aspect of the model. We have contextualised our study within the framework for organic–clay nanocomposites developed by Kennedy et al. (2002, *Science*), Kennedy et al. (2014, *EPSL*), Keil and Mayer (2014, *Treatise on Geochemistry*), Bennet et al. (2012, *Marine Geology*), Mayer (1994, *GCA*), Ahn et al. (1999, *American Mineralogist*) and Ransom et al. (1997, *Marine Geology*).

As noted by Curry et al. (2007, *GCA*), "Future research opportunities could investigate the preservation of OM in relationship to fabric changes and sediment diagenesis during deep

burial and large overburden consolidation.” We believe that our study addresses this challenge, which is not given attention in the previous works cited above due to the nature of the rocks studied (younger, relatively unmetamorphosed rocks). We show, at the nanoscale, how clay minerals deposited in not dissimilar environments more than 2.9 Gyr ago can undergo phase transformation and contribute to the retention of primary biogeochemistry. This reflects a novel contribution and extends the scope of the excellent work in the above-cited studied to more ancient sedimentary rock sequences.

We strongly suspect that the samples we have studied would have been deposited as materials akin to those described in Ransom et al. (1997, *Marine Geology*), evolving through the types of materials described in Kennedy et al. (2014, *EPSL*) to become subgreenschist-like materials similar to those in Ahn et al. (1999, *American Mineralogist*). We have integrated some discussion of this evolutive aspect into the discussion.

We have added substantial additional text to the discussion in three places, namely the paragraphs starting with:

- “Following deposition, the preserved associations between phyllosilicates and well-preserved organic materials...”
- “The graphitisation of organic materials in geological sequences is influenced by a number of factors...”
- “Well-preserved associations of phyllosilicates and organic materials have been recognised for decades and direct observations...”

We have also slightly modified the abstract to explicitly account for the novelty of this contribution: “However, the precise micromechanics by which exceptional retention of organic microbial traces occur within such rocks **over billion-year geological timescales** remain poorly understood.”

Preservation Mechanism. The study suggests that authigenic phyllosilicates inhibit the carbonization of organic materials by "pressure compensation" linked to the kaolinite–illite transformation. This is plausible, but other factors, such as the low thermal conductivity of clay-rich layers, may also contribute to preserving organic material by creating localized, cooler microenvironments within phyllosilicate-rich zones. Such conditions could reduce thermal degradation in phyllosilicate-rich layers relative to the more porous, thermally conductive sandstone layers.

Suggestion: Since burial and metamorphism likely affected the entire stratigraphic section (sandstones and phyllosilicates), and the only temperature reading (350°C) was from organic-rich phyllosilicate layers, consider estimating the temperature of the organic-poor microbial mats within sandstones (e.g., Fig. S2). While the organic content in these sandstones is considerably lower, obtaining a single temperature reading from organics that are less shielded by phyllosilicates could allow for a comparison and clarify any potential thermal and maturity disparity.

Thank you very much for the recommendation to perform these additional data treatments. As suggested, we have analysed Raman geothermometry data using rare organic materials in the microbial mat-poor sandstone matrix, and compared this with Raman data from the microbial mats themselves. This shows a slight disparity in peak thermal history of approximately 0–30°C, depending on sample, which supports our hypothesis of differential degrees of maturation – and thus preservation – in sandstone layers (‘background’ deposition) *versus* layers dominated by phyllosilicates and organic materials. We have framed these findings in terms of thermal conductivity, thermal degradation and porosity as highlighted in your comment above. The results obtained and their associated errors (inherent to the geothermometers used) are reported in Table S1 in the Supplementary Information.

Suggestion: since phyllosilicates are common across various geological settings, it would be valuable to assess whether the MCF phyllosilicates exhibit specific chemical or physical characteristics that might enhance preservation. Are the MCF phyllosilicates or the illitization processes during diagenesis distinct from those in other formations, or might these mechanisms apply more broadly to siliciclastic environments worldwide?

Thank you for this suggestion. Unfortunately, since no work has been done at a similarly high resolution on microbial fossil-bearing horizons of similar maturity from other Archaean sequences, the data to address this question do not yet exist. We are currently working on similar analyses of the 3.47 Ga Middle Marker horizon of the Barberton greenstone belt and hope to evaluate this question using the data obtained. Similar analyses of the 3.22 Moodies Group (also Barberton greenstone belt) would like enable similar evaluations. We have included an indication of this as an important future research direction in the discussion:

“Future research should consider other phyllosilicate-rich environments, particularly those with distinct mineralogies but similar thermal maturities and metamorphic histories, to evaluate the potential universality of the mechanism proposed over billion-year geological timescales. Such studies may unveil further novel taphonomic windows, thereby extending the known range of biosignature repositories in Earth’s most ancient rocks.”

Minor Comments:

•Line 45, 340, 396, 421: The term ‘graphitization’ is typically associated with temperatures >650°C. A possibility would be to use the term ‘maturation’ in some cases (lines mentioned) instead.

We have used the term “maturation” where recommended.

•Line 63: consider adding Noffke et al., 2002 and 2006 papers to the 2022 reference, as it is important to mention that fossilized microbial biomass has been described in coarser-grained siliciclastic sedimentary rocks for decades.

We have added the recommended references; thank you for identifying these.

•Line 101 and Supplementary Data (Geological setting): The terms "organic-rich" and "organic-poor" microbial mats could benefit from quantification. Is there a Total Organic Carbon (TOC) measurement available?

Unfortunately, bulk TOC data are not available. We deliberately focused the present manuscript on *in situ* analyses. Bulk TOC from samples significantly larger in volume than the regions analysed might not be representative of the microscopic zones of interest. We based our distinction of organic-rich and organic-poor on the results of optical and electron microscopy and Raman and FTIR microspectroscopy. To mitigate any confusion, we have reworded to “relatively organic-rich layers of the samples” and “relatively organic-poor layers of the samples” where appropriate.

•Line 202: The use of terms like "poorly ordered, partially graphitised turbostratic carbon" is appropriate, but it would be helpful to include a brief explanation for readers, particularly for those who may not be familiar with the significance of this type of kerogen in relation to microbial preservation who may not be familiar with these specific terms.

Thank you for noting this – we agree and have provided brief definitions to define these types of terms: “...the MCF microbial mats are preserved as kerogen with amorphous (i.e. without long-range crystalline structure) to poorly ordered, partially graphitised turbostratic (i.e.

partially ordered regions of graphene clusters with weak long-range crystalline structure) retaining an unprecedented level of primary biogeochemical heterogeneity.”

•Line 300: The manuscript mentions extracellular polymeric substances (EPS) as the dominant constituent in the study microbial mats. Given the known resilience of EPS to diagenesis, this is a plausible hypothesis. Are there specific molecular markers or features in the spectroscopic data that point explicitly to EPS preservation, rather than just bulk organic material?

In our experience with modern and fossilised geobiological samples, EPS forms a major part of the volume of the organic materials within such ecosystems and we therefore consider it logical to anticipate that much of the bulk organic materials derive from such sources. As you also note, EPS is particularly resilient to diagenesis (we have observed this in many of our own samples). We find it challenging to ascribe specific molecules or spectral features to EPS over bulk kerogen as many of the same organic groups will be present in each, but have highlighted several possibilities: “it is therefore anticipated that a portion of the biogeochemical complexity in these fossilised microbial mats arises from preserved EPS, potentially some fraction of the aromatic, aliphatic and carboxyl groups identified in FTIR and STXM data.”

•Line 317: Compare $\delta^{13}\text{C}$ fractionation values to known microbial metabolic pathways to contextualize these results.

Following comments from the other reviewers, we have elected to remove the carbon isotope fractionations from this manuscript. All reviewers have requested further details on this point, and answering these comments satisfactorily would require the addition of a separate section to the discussion, which would be secondary to the main purpose of this manuscript – to describe the nanomechanical processes promoting exceptional organic preservation in these samples. We do not wish this manuscript to become a discussion of the utility of carbon isotopes on ancient samples, and furthermore, we were only able to acquire a small number of $\delta^{13}\text{C}$ measurements (due to the time-consuming nature of the NanoSIMS analyses conducted), hence the discussion would be inherently incomplete. We are currently working to obtain coupled carbon, nitrogen and sulphur isotope characterisation of a larger number of Mosquito Creek Formation samples for a future manuscript.

•Line 365: If primary productivity centers around anoxygenic photosynthesis (indicating anoxic conditions) (line 329), wouldn't stage 3 of the depositional model be implicit?

We apologise for any ambiguous wording at this point. Stage 3 of the model includes the covering of the mats by new sedimentary layers, but “anoxic” here was intended to imply that the problem of oxidation (and flushing out) of organic matter as expected in high-energy depositional conditions does not pose an issue in this context. We have reworded this to “Stage 3: development of isolated anoxic microenvironments” and hope that this more clearly explains this stage of the model. The phenomenon of isolation of organic domains is now also referenced more clearly elsewhere in the manuscript.

•Line 714: Equation 1 is from Kouketsu et al. (2014) and Equation 2 is from Beyssac et al. (2002).

Thank you for noting this oversight. These have been corrected.

Reviewer #3

This paper presents a detailed study of organic remnants of ancient life in a coarse-grained siliciclastic sedimentary rock. As the authors point out, such rocks have often been ignored, because it is generally assumed that traces of life are poorly preserved within them. This type of sediment represents an energetic depositional environment where low-density organic materials are not easily retained. Also, during diagenesis these rocks would be easily be penetrated by oxidative fluids. However, some intriguing observations suggest that there are mechanisms for preservation in such rocks.

The current paper presents a very detailed study of exactly such a mechanism. The authors studied drill core samples of a sandstone from the 2.97 Ga Mosquito Creek Formation in Australia. They discovered organic layers associated with aluminous phyllosilicates. First they used an impressive array of high-resolution techniques to structurally, chemically and isotopically characterize these organic layers. Then they focused on the aluminosilicates themselves, and established that kaolinite-to-illite conversion during diagenesis is one of the key mechanisms for the preservation of the organic materials. This illitization caused volume reduction of the rock matrix, preventing extensive graphitization of the organic fractions. It also prevented fluid circulation through pore spaces.

The authors describe in detail how this mechanism would have led to pockets of well-preserved microbial mats. This study thus greatly expands the possibilities of finding traces of life in yet another type of Archean rocks.

I think this is an important study that should be published. However, there are some issues that require minor to moderate revisions. My comments below can hopefully be used to strengthen the manuscript.

Thank you very much for your positive and highly constructive review of our manuscript. We have provided the additional data and discussion requested in these comments and the revised version of the manuscript.

Comments:

1) Interpretation of the organic layers:

The organic layers are described in detail in Lines 131-151. These descriptions are key for the interpretation that these represent biofilms/microbial mats. I think there are some shortcomings here. Some of the descriptions in the text are difficult to see in the figures, and it is not adequately discussed how these observations exclude alternative explanations, particularly the possibility that some of the observed organic layers may represent veins of organic fluids.

Important observations in the text here are the following. First there is the mention that the layers bifurcate/anastomose around sediment grains. (Fig.1B-C). Also, in line 138-139 it is said that the layers deform beneath overlying sediment particles and exhibit local rip-ups, tear-ups and crumpling. And in line 140 it is said that sedimentary particles are oriented parallel to laminae (Fig.1D). I carefully looked at these figures 1A-D, and I have a hard time seeing what is described here. This likely has to do with the small size of the images in Fig.1, and lack of annotations. The figures S2-S6 are more detailed, but it would be good if it can be specifically pointed out where in these images the key observations are made that are described in this part of the text (line 131-151).

We appreciate the need for further detail and annotations in the figures. As you note, more detailed characterisation was provided in Figures S2–S6; therefore, we have separated Figure 1 into two separate figures, the new Figure 1 presenting annotated optical microscopy images

(including some images from Figs. S2–S6) and Figure 2 presenting the microspectroscopy analyses. As requested, we have carefully tied the optical microscopy observations to the descriptions in the text, and have indicated at least one example of each bio-indicative feature described in the figures, including 1) undulatory laminations; 2) bifurcation and anastomosis; 3) entrainment of sedimentary particles; 4) rare rip-up/roll-up structures; and 5) microbial mats mantling underlying sedimentary layers.

We hope that the addition of these panels to Figure 1, coupled with the additional discussion of non-biological alternatives in the accompanying text (see also response below) fully explains our interpretation of these microstructures.

Also, I miss zoomed-out images of the actual extent of the layering. Only pictures of the thin sections are shown. Are there images of the cores themselves? It should be possible for a reader to see these important observations in a figure in the main paper itself.

No other cores were taken through the Mosquito Creek Formation as part of the Achaean Biosphere Drilling and it is therefore impossible to know the true extent of the mats. In light of similar observations that have been made using the BASE (Barberton Archaean Surface Environments) cores taken through siliciclastic sediments in the 3.22 Ga Moodies Group, one might assume a similar extent of up to tens of kilometres laterally, although preservational conditions may also vary. Unfortunately, we were not provided with images of the core samples.

In line 147-150 it is written: "On the basis of their sedimentary context, organic composition and morphological identification to known similar structures from throughout the geological record (refs...) we interpret these laminated structures as fossilized siliciclastic microbial mats. After this sentence, I miss a bit more discussion on alternative explanations for the observed organic layers. The authors quickly jump to the conclusion that these represent microbial mats, without discussing the possibility that biogenic organic fluids could have migrated layer-parallel through the rock. Have these rocks been influenced by hydrothermal fluids? Can this effectively be excluded that these represent layer-parallel organic veins? I'm confident that the authors can establish this. I'm mostly asking for more clarity, and possibly some annotations in the figures.

Thank you for encouraging us to expand upon this. We hope that the additional annotations in the revised Figure 1 outline most of our reasoning for the interpretation of these features as biogenic microbial mats. As recommended, we have also included an evaluation of the potential non-biological processes that may account for such features. In brief:

- At the scales of observation reported herein, the organic materials bear no characteristics resembling the nanoporous pyrobitumen reported in oil-saturated cherts (e.g. Rasmussen et al., 2021, *Astrobiology*); in particular, there is no petrographic evidence for the emplaced and thermally altered (coked) pyrobitumen intergrown with polymetallic sulfides, which has been shown to be diagnostic of such hydrocarbon migration processes in ancient rocks (Rasmussen and Buick, 2000, *Geology*).
- Throughout the entirety of the Mosquito Creek Formation (Ohmoto et al., 2006, *Science*; Bagas et al., 2008, *Precambrian Research*), there is no evidence for carbon-rich sediments that formed during precipitation from silica-rich, carbon-bearing hydrothermal fluids in vein systems and vent-proximal seafloor sediments (*cf.* Rasmussen et al., 2023, *Science Advances*).
- No geochemical evidence (e.g. strongly positive Eu anomalies > 2 , general trace metal and REE enrichment) exists in these sequences to suggest that hydrothermal processes have played a leading role in the generation of organic carbon (Bagas et al., 2008).

2) Raman spectra of the organic phases:

The use of the Raman-based geothermometers is unclear.

In line 156-159 it is established, based on Raman spectroscopic analysis, that the carbonaceous phases experienced a peak metamorphic temperature of 343-348 C. This is extremely precise, and apparently was based on two geothermometers, Beyssac et al. (2002) and Kouketsu et al. (2014). It would be amazing if these two different geothermometers would reach the same temperature with such a small error margin.

I tried to find the details of the calculations made by the authors, and the only thing I found was a mention of the geothermometers in the Methods section, lines 715-716. In fact, the geothermometer of Beyssac et al. (2002) is incorrect as it is listed there for two reasons: 1) Apparently peak-intensities were used, while peak-areas should have been used, 2) Only the D1 and G-peaks were used, while the D2-peak (shown in the spectrum in Fig.S23) was ignored.

So the authors should list the correct formula (as it is given in Beyssac et al., 2002) for this geothermometer in line 716, and they should show (maybe in the supplement) how exactly the calculation was made for this particular geothermometer.

Then, for the Kouketsu geothermometer, the correct formula is used, but it should be shown what the exact temperature is that is obtained with this particular geothermometer. It would be very nice if indeed this gives the same temperature as the one of Beyssac et al. (2002), but this should be verifiable.

We apologise for the oversight which included the incorrect Raman geothermometer for Beyssac et al. (2002). This has been corrected as $T(^{\circ}\text{C}) = -445(R2) + 641$. The original calculations (using sample 28c) were made using the correct geothermometer and the D2 peak; this was simply an oversight in reporting in the manuscript.

In response to your comments and those of another reviewer, we have recalculated a range of Raman geothermometry results from each of samples 28c, 26d, 26c and 21b (highlighted elsewhere in the manuscript). As you suggested, this has led to a larger range of values (313–363°C, with one potentially spurious value of 285°C), although we note that the errors associated with the Beyssac and Kouketsu geothermometers are 50°C and 30°C, respectively. With values that fall within a ~40°C range, we consider that the two Raman geothermometers are mutually supportive.

We have provided the details of the spectral deconvolution (i.e. the parameters used in the calculations) in the supplementary materials as requested.

3) FTIR data:

In lines 191-199 the results of FTIR micro spectroscopy are presented. Various important organic bonds and groups are found (aromatics, aliphatics, etc.). However, there is no reference to any figure here. The only thing I found were the FTIR maps in Fig.1E. It would be good to see the FTIR spectra from which we can actually determine whether aromatic and aliphatic groups are present. Also, in line 1970-199 it is said that based on averaging 30 spectra, the CH₃/CH₂ ratio in the organic layers could be established. How was this done? And again, where are the spectra?

We appreciate the need for further clarity regarding these data.

We have provided a selection of representative spectra in the manuscript for several samples in Figure 3, indicating the positions of the most important organic bonds and groups identified, focusing on the aromatic and aliphatic groups. We also specify a number of closely associated and overlapping peaks in the 1300–1800 cm^{-1} region arising from the silica matrix. The distributions of these groups are visually represented in the maps, which we have retained in Figure 2.

We also acknowledge the need for further detail regarding the calculation of the CH_3/CH_2 ratios. This was performed using the ThermoScientific OMNIC Picta software, through which we determined the areas beneath the C–H peaks for 30 high signal:noise ratio spectra following background subtraction from multiple samples. To account for the inherent uncertainty in these values (influenced by the background subtraction applied) and to better account for the range of values observed, we have presented a more cautious version of this description in the revised manuscript, highlighting the full range of values calculated using the OMNIC Picta software (0.1–1.1) and the most common values (~ 0.4 – 0.5).

4) NanoSIMS d^{13}C analysis:

The carbon isotope analysis by NanoSIMS is poorly described. Nothing is said in the Methods section about the carbon isotope standards that were used, nor of the count rates that would constitute a significant signal above background. Also, what is the error on the analysis? With NanoSIMS this can easily be 5 per mil or more. It is important for a reader to be able to independently evaluate the quality of the data.

The NanoSIMS maps in Figure S37 only show maximum counts of ions. It is very difficult to derive a meaningful d^{13}C value from this.

In the main text, in lines 252-253 the d^{13}C values of the organic layers are presented. The range of $\text{d}^{13}\text{C} = -35$ to -63 per mil. This lower limit is extremely low. Can the authors verify that this number is correct?

5) Interpretation of metabolism, based on d^{13}C

In the Discussion, lines 317-329, the authors describe a range of metabolisms that could be responsible for the observed d^{13}C range. These include anoxygenic photosynthesis, methanogens, methanotrophs, sulphate-reducing pathways, and anaerobic nitrogen fixation. That is quite a list, and again a $\text{d}^{13}\text{C} = -63$ per mil is really extremely low. In my opinion such a low number would specifically suggest methanotrophy.

However, at the end of this section, in line 328-239, the authors draw the conclusion: "Coupling stratigraphic context, inorganic geochemistry and d^{13}C measurements suggests that primary productivity in the mats was likely centered on anoxygenic photosynthesis."

How can this conclusion actually be drawn, based on the available information? I would think it is quite difficult to narrow down such a list of possible metabolisms to a single one. I think this is over-interpretation. In any case, it needs to be better explained why this particular metabolism can be inferred.

In response to your points 4 and 5 above, and following similar comments from the other reviewers, we have elected to remove the carbon isotope fractionations from this manuscript. All reviewers have justifiably requested further details on this point, and answering these comments satisfactorily would require the addition of a very detailed separate section in the discussion, which would be secondary to the main purpose of this manuscript – to describe the nanomechanical processes promoting exceptional organic preservation in these samples. We believe that this addition would severely and negatively redirect the focus of the

manuscript to be on metabolic reconstruction of the microbial ecosystem, simultaneously detracting from its novelty. Because you and the other reviewers have highlighted that the main strength of this manuscript is in the understanding of organic taphonomy, we prefer to eliminate this weakness and focus on the strength of our contribution.

We do not wish this manuscript to become a discussion of the utility of carbon isotopes on ancient samples, and furthermore, we were only able to acquire a small number of $\delta^{13}\text{C}$ measurements (due to the time-consuming nature of the NanoSIMS analyses conducted), hence any discussion section added to address this would be inherently incomplete – as you note, it is difficult to narrow down to a single metabolism based only on carbon isotopes. We are currently working to obtain coupled carbon, nitrogen and sulphur isotope characterisation of a larger number of Mosquito Creek Formation samples for a future manuscript.

Thank you again for your highly constructive review of our manuscript and we hope that our responses to your comments and the improvements made as a result of your recommendations satisfy any remaining concerns.